# EMBO *reports*

# Insulin and epidermal signals independently shape sexually dimorphic neurite branching in *C. elegans*

Jia-Bin Yang [1], Rui-Tsung Chen[1], Yun-Yu Chen[1], Yun-Hsien Lin[1] & Chun-Hao Chen [1,2]✉

## Abstract

**Sexual dimorphism in neural wiring and behavior arises from both intrinsic genetic programs and environmental cues, yet how these factors interact to shape neuronal morphogenesis remains unclear. Here, we investigate sexually dimorphic collateral branching in PVP cholinergic interneurons of *Caenorhabditis elegans*. In hermaphrodites, PVP branches form near the vulva and exhibit dynamic morphologies enriched with synaptic proteins for dense core vesicles but not synaptic vesicles, suggesting a role in selective neuropeptide transmission. We find that sex identity is necessary but not sufficient for PVP branching. Sex identity engages autonomous insulin signaling via the FOXO transcription factor DAF-16 to promote branch formation and modulate dynamic branch morphologies according to nutritional status. However, external epithelial cues from primary vulval cells are both necessary and sufficient to induce branching independent of sex identity. Despite acting through distinct pathways, insulin signaling and vulval cues converge on F-actin cytoskeletal remodeling. These sexually dimorphic PVP branches modulate egg-laying behavior in hermaphrodites. Our study uncovers a multilayered regulatory framework integrating intrinsic sex-specific programs and extrinsic signaling to shape sexually dimorphic neural circuits.**

**Keywords** Sexual Dimorphism; Neuronal Branching; Morphological Plasticity; PVP Neurons; *Caenorhabditis elegans*
**Subject Categories** Neuroscience; Signal Transduction

## Introduction

Biological sex profoundly influences neural processing and behavior by establishing sexually dimorphic neural circuits during neurodevelopment (Barr et al, 2018; Jarrell et al, 2012; Sato and Yamamoto, 2023). Typically, this dimorphism arises from transcriptional programs that specify sex-specific characteristics within the nervous system (McPherson and Chenoweth, 2012; Robinett et al, 2010; Serrano-Saiz et al, 2017). In vertebrates, these transcriptional programs predominantly control gonadal hormone production, broadly influencing neuronal identity and connectivity

(McPherson and Chenoweth, 2012). However, the extensive effects of hormones on neurons and surrounding cells complicate efforts to isolate the specific contributions of individual cells to sexually dimorphic neural development.

In contrast, invertebrates such as *Drosophila melanogaster* and *Caenorhabditis elegans* (*C. elegans*) provide unique advantages for studying cell-autonomous mechanisms of sex determination. In these organisms, sex is determined independently within individual cells, facilitating a more precise delineation of intrinsic cellular contributions. Studies have demonstrated that cell-autonomous transcriptional programs are essential for establishing sexually dimorphic neuronal connectivity (Robinett et al, 2010; Serrano-Saiz et al, 2017). Additionally, sex can influence the production of environmental cues that modulate neural wiring (Oren-Suissa et al, 2016; Weinberg et al, 2018), highlighting the complex interplay between intrinsic sexual identity and external factors. Consequently, a significant question remains about how intrinsic and extrinsic factors interact to shape sexually dimorphic neural networks.

*C. elegans*, a hermaphroditic species with males and self-fertilizing hermaphrodites, has a nervous system comprising sex-shared and sex-specific neurons (Barr et al, 2018). While most sex-shared neurons exhibit similar morphologies, a subset of sensory and motor neurons, including PHC, DVB, PDB, and PVD neurons, shows sexually dimorphic morphology that results in sex-specific connectivity and behaviors (Hart and Hobert, 2018; Iosilevskii et al, 2025; Kim et al, 2025; Serrano-Saiz et al, 2017). Hermaphroditic fate is controlled by the Gli-type zinc finger transcription factor TRA-1, while TRA-1 degradation via male-specific FEM-3 determines male fate (Hunter and Wood, 1990; Starostina et al, 2007). These intrinsic transcriptional programs are well known to direct neurodevelopment through processes such as proliferation, apoptosis, synaptic pruning, and trans-differentiation (Garriga et al, 1993; Molina-Garcia et al, 2020; Salzberg et al, 2020; Serrano-Saiz et al, 2017; Shen and Hodgkin, 1988). However, environmental cues also significantly shape sexually dimorphic connectivity. For example, sex-specific synaptic pruning of chemosensory PHA neurons in *C. elegans* is reversible upon sexual transformation of synaptic partners, and male-specific synaptic connections formed by DVB motor neurons depend on environmental signals (Hart and Hobert, 2018; Oren-Suissa et al, 2016; Peedikayil-Kurien et al, 2024). Moreover, peripheral tissues have been shown to promote male-specific muscle and gonadal development via non-autonomous mechanisms by secreted factors in *Drosophila*

[1]Institute of Molecular and Cellular Biology, College of Life Science, National Taiwan University. No. 1, Sec. 4, Roosevelt Rd., Taipei 10617, Taiwan. [2]Department of Life Science, College of Life Science, National Taiwan University. No. 1, Sec. 4, Roosevelt Rd., Taipei 10617, Taiwan. ✉E-mail: chunhaochen@ntu.edu.tw

(DeFalco et al, 2008; Lawrence and Johnston, 1986). These findings highlight a complex interplay between autonomous and non-autonomous factors in sexually dimorphic neurodevelopment, although the molecular details underlying this interaction remain poorly understood.

Sexually dimorphic morphogenesis ultimately manifests as structural changes in neurons, critically dependent on cytoskeletal proteins. Our previous studies indicate that dynamic F-actin is integral to multiple phases of neurite branching. During neuronal development, local enrichment of F-actin precedes neurite branching sites mediated by the Wnt morphogen gradient (Chen et al, 2017). In addition, F-actin can form uniform waves in growth cones, promoting branching on fasciculated neurites (Chen et al, 2019). In mature neurites, F-actin retains its dynamic properties, which are essential for synaptic remodeling (Colicos et al, 2001). However, the specific regulation of F-actin dynamics within sexually dimorphic neuronal structures remains elusive.

In this study, we focused on the sexually dimorphic development of collateral branching in the sex-shared PVP cholinergic interneurons of *C. elegans*. In hermaphrodites, PVP neurons formed collateral branches near the vulva during adulthood, a feature absent in males. This dimorphism emerged during sexual maturation and was marked by F-actin accumulation. Mature branches were enriched with dense-core vesicles but lacked synaptic vesicles, suggesting a role in neuropeptide release. In addition, mature branches also accommodated cilia proteins, although their functional role remains uncertain. Mechanistically, we demonstrated that intrinsic sexual identity and environmental stimuli jointly modulated the F-actin cytoskeleton via the insulin signaling pathway and interactions with vulval epithelial cells. Furthermore, the transcription factor DAF-16, a component of the insulin pathway, adjusted branch morphology in response to nutritional cues, highlighting the adaptive plasticity of PVP branches in hermaphrodites. Loss of PVP branches led to impaired egg-laying behaviors, suggesting the physiological significance of sexually dimorphic PVP branching in hermaphrodites. These findings reveal a sophisticated interplay between autonomous and non-autonomous factors in shaping sexually dimorphic neural connectivity.

# Results

## PVP interneurons exhibit sexually dimorphic branching in hermaphrodites

PVP neurons are cholinergic interneurons in the tail extending their processes from the pre-anal ganglion to the nerve ring (Fig. 1A) (Pereira et al, 2015; White et al, 1986). During early embryogenesis, PVP neurons act as pioneers to guide neurite projections along the ventral nerve cord through neurite fasciculation (Wadsworth et al, 1996; Wadsworth and Hedgecock, 1996). In adult hermaphrodites, electron microscopy and fluorescent reporter analyses have revealed a branching structure near the vulva, although the molecular mechanisms underlying PVP branching remain unknown (Christie and Koelle, 2022; White et al, 1986).

To study PVP branching, we used the 0.5 kb *ocr-3* promoter to drive the expression of NeonGreen to visualize PVP neurons (Lorenzo et al, 2020). *pdf-1* encodes a neuropeptide secreted by multiple neurons in the head and PVP neurons in the tail (Flavell et al, 2013). The transgenic strain carrying *Pocr-3::NeonGreen*

specifically labeled PVP neurons, confirmed by co-expression of a transcriptional reporter driven by the *pdf-1* promoter (Fig. EV1A). Consistent with previous studies, collateral branches were observed in hermaphrodites near the vulva at the adult stage (Fig. 1A',B) (Christie and Koelle, 2022; White et al, 1986). We also confirmed that multiple copies of transgenes generated via traditional germline transformation diminished PVP branching in a dose-dependent manner, suggesting that high transgene copy numbers interfere with branch formation in PVP neurons (Fig. EV1B) (Christie and Koelle, 2022). To address this issue, we employed CRISPR-Cas9 to insert a single copy of *Pocr-3::NeonGreen* into chromosome I (Fig. EV1C; see Methods). Using this single-copy transgenic strain, we observed robust PVP branch formation in nearly all adult hermaphrodites, validating its use as a reporter for studying PVP branching development (Fig. EV1B). Therefore, we used the reporter strain to investigate the development of PVP branches in this study.

Remarkably, we found that PVP branching was absent in males. Only a few males exhibited short and disoriented protrusions (Fig. 1A"). In hermaphrodites, PVP branches consistently projected dorsally near the vulva with two types of branch morphologies, wing- and rod-like structures (Fig. 1A",B,D,E). In contrast, protrusions in males were randomly distributed with a slight bias toward the head region (Fig. 1A,A"). These findings indicate that PVP branching is sexually dimorphic between hermaphrodites and males.

## Dense-core vesicles are selectively enriched in sexually dimorphic PVP branches

To further investigate the development of PVP branching in hermaphrodites, we conducted time-course experiments. We observed that PVP branches began to emerge during the L3 larval stage and were fully developed near the vulva by the adult stage (Fig. 1C). These branches displayed axonal characteristics, as evidenced by the presence of presynaptic proteins expressed under the *ocr-3* promoter. In particular, the transmembrane protein IDA-1, a marker of dense-core secretory vesicles, was strongly enriched in PVP branches, whereas CLA-1, an active zone protein required for synaptic release, was only sparsely detected (Xuan et al, 2017; Zahn et al, 2004). However, synaptic vesicles labeled by the small GTPase RAB-3 were notably absent from the branches but were enriched in the nerve ring, suggesting a selective role for PVP branches in mediating peptidergic signaling (Fig. 1D; Appendix Fig. S1A). This selectivity did not extend to cellular organelles, as mitochondria and the trans-Golgi network, marked by TOMM-20 and TGN-38, respectively, were distributed throughout the processes and branches (Appendix Fig. S1B). In contrast to hermaphrodites, we did not detect the CLA-1 marker in short and disoriented protrusions in males.

While our initial characterization suggested axonal features, a previous study also suggests that wing- and rod-like structures of PVP branches resemble dendritic structures in sensory neurons (Christie and Koelle, 2022). To investigate whether these wing- and rod-like structures possess dendritic features, we analyzed single-cell RNA sequencing data from CeNGEN and identified genes associated with cilia function (Taylor et al, 2021). We found that *kap-1*, which encodes a component of the kinesin complex involved in intraflagellar transport, is expressed in PVP neurons. KAP-1 labeling revealed strong enrichment in the cell soma and the branch

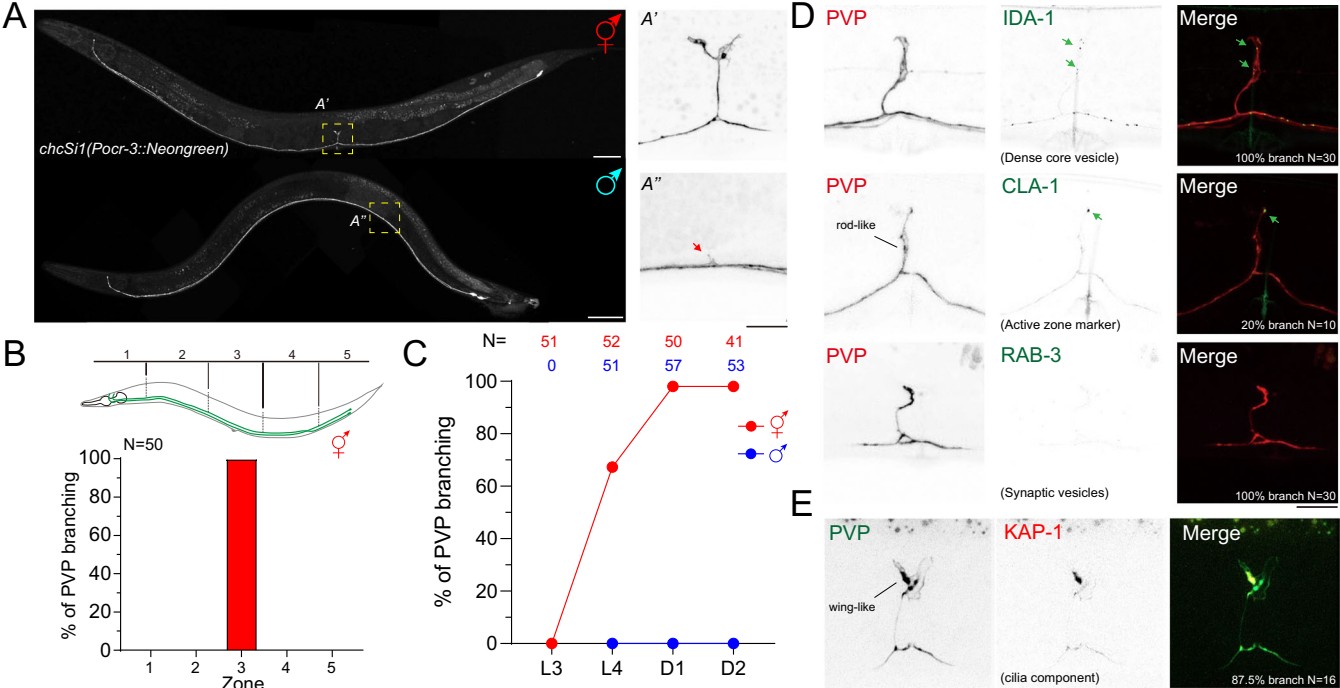

**Figure 1. PVP cholinergic interneurons form collateral branches in adult hermaphrodites.**

(A) Z-projection of confocal fluorescent images for PVP neurons in hermaphrodites and males labeled by *chcSi1(Pocr-3::NeonGreen)*. Scale bar = 50 μm (hermaphrodites) and Scale bar = 20 μm (males). (A', A'') enlarged confocal images of PVP branches in hermaphrodite and male, respectively. The arrow indicates a protrusion of male PVP neurons. Scale bar = 10 μm. (B) Histogram of PVP branch location and PVP branching in hermaphrodites. *N* number is indicated. (C) Quantification of sexually dimorphic PVP branching at the indicated developmental stages. *N* numbers are indicated. (D) Z-projection of confocal fluorescent images of marker labeling dense-core vesicles (IDA-1), active zone proteins (CLA-1), and synaptic vesicles (RAB-3) in *chcEx104[Pocr-3::IDA-1::GFP]*, *chcEx034[Pocr-3::GFP::CLA-1]*, and *chcEx032[Pocr-3::mCherry::RAB-3]* transgenic hermaphrodites, respectively. Scale bar = 10 μm. The percentage of animals carrying each marker was shown in the merged figure. The average puncta number for IDA-1 and CLA-1 is 4.875 (± 3.41, *n* = 32) and 1 (± 0, *n* = 2), respectively. Arrows indicate punctae signal in PVP branches. (E) Z-projection epifluorescent images of markers labeling KAP-1 by *chcEx251[Pocr-3::KAP-1::mCherry]*. Scale bar = 10 μm. The percentage of animals carrying the KAP-1 marker was shown in the merged figure. *N* indicates the number of biological repeats in this figure. Source data are available online for this figure.

tip (Fig. 1E; Appendix Fig. S1C). Notably, Christie and Koelle (2022) did not detect other ciliary markers in PVP branches, leaving the biological function of KAP-1 in these structures unresolved (Christie and Koelle, 2022). Nevertheless, these findings indicate that PVP neurons exhibit sexually dimorphic morphogenesis, acquiring distinct structural and functional features during sexual maturation.

## Enrichment of cytoskeleton proteins precedes PVP branching

The enrichment of cytoskeletal proteins is a hallmark of early neurite branching events (Bilimoria and Bonni, 2013; Chen et al, 2017; Chia et al, 2014). To investigate this process, we examined the dynamics of F-actin using the LifeAct marker driven from the *ocr-3* promoter (Riedl et al, 2008). Before branch formation at the L3 larval stage, F-actin was predominantly enriched in the cell soma and nerve ring (Appendix Fig. S2). As development progressed, F-actin began forming scattered patches along the neuronal process, which became increasingly concentrated in the vulva region (Fig. 2A–C). By the late L4 stage, F-actin was highly enriched in small protrusions and ultimately localized within the newly formed branches (Fig. 2D).

In adult hermaphrodites, F-actin organization differed based on branch morphology. Wing-like branches showed concentrated F-actin at protrusions and terminal ruffles, while rod-like branches exhibited a uniform F-actin distribution along the shaft (Fig. 2D,E). To explore how these two branch structures form, we performed live-cell imaging of mature PVP branches. Remarkably, we observed that rod-like branches expanded their membranous structure to become wing-like branches and vice versa, demonstrating dynamic interchanges between the two morphologies (Fig. 2F). These findings suggest that F-actin plays a multifaceted role that contributes to the early stages of branch development as well as the structural shaping of mature branches.

## PVP branch morphology adapts to nutritional status

Nutritional status is known to modulate gene expression and neuronal morphology in *C. elegans* (Ryan et al, 2014). To determine whether the transition between wing-like and rod-like PVP branches is influenced by nutritional status, we subjected adult hermaphrodites to dietary restriction by feeding them diluted bacterial diets ($10^{-3}$ or $10^{-6}$) for 16–18 h. While the dietary restriction of adult hermaphrodites did not affect PVP branching (Fig. EV2A), we observed a dose-dependent decline in wing-like branches, accompanied by an increase in rod-like

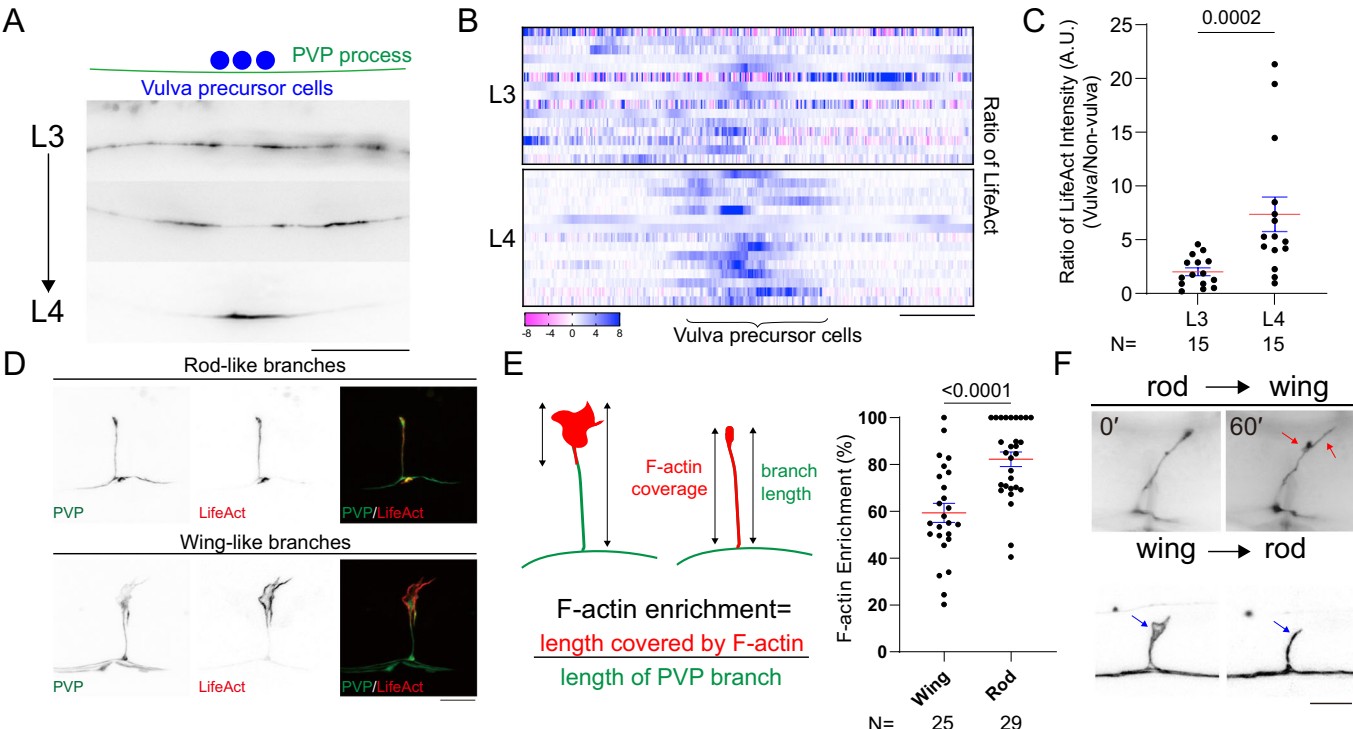

**Figure 2. F-actin assembly occurs during PVP branching and differentially distributes to wing- and rod-like branches.**

(A) Fluorescent images of LifeAct (labeling F-actin via *chcEx070[Pocr-3::LifeAct::mKate]*) in PVP neurons at the indicated developmental stages. Scale bar = 20 μm. (B) Heatmap of the ratio of LifeAct in the PVP process at different developmental stages. N = 15 animals. Scale bar = 5 μm. Anterior is to the left. (C) Quantification of the ratio of LifeAct average intensity of the vulva and non-vulva region at the indicated developmental stages. Mann–Whitney U-test. Error bar indicates SEM. N number and P value are indicated. (D) Z-projection of confocal fluorescent images of LifeAct in the wing- and rod-like branches. Scale bar = 10 μm. (E) (left) A schematic diagram showing the quantification of F-actin distribution in PVP branches. (right) Quantification of F-actin distribution in wing- and rod-like branches. Student t-test. The error bar indicates SEM N number and P value are indicated. (F) Z-projection epifluorescent images of PVP branches with wing-to-rod or rod-to-wing transition. Red arrows indicate the expansion of a rod-like PVP branch and blue arrows indicate the shrinkage of a wing-like PVP branch. Scale bar = 10 μm. N indicates the number of biological repeats in this figure. Source data are available online for this figure.

branches (Figs. 3A and EV2B). Notably, these morphological changes were reversible, as re-feeding with undiluted diets for 16–18 h restored the wing-like branches (Fig. 3A). These findings demonstrate that PVP branch morphology is highly dynamic and closely tied to physiological conditions, highlighting a reversible and nutritionally regulated plasticity in neuronal structure.

## The DAF-2/DAF-16 insulin pathway autonomously promotes PVP branch formation

Nutritional status is primarily sensed through the insulin signaling pathway, which involves the DAF-2 insulin-like receptor and the DAF-16/FOXO transcription factor in *C. elegans* (Gottlieb and Ruvkun, 1994). Under starvation conditions, reduced DAF-2 activity allows DAF-16 to translocate into the nucleus, promoting the expression of stress response and dauer-related genes (Henderson and Johnson, 2001). To study the role of this pathway in PVP branching, we examined *daf-2* and *daf-16* mutants. Loss of *daf-2* significantly reduced PVP branching, while the *daf-16* mutation had a limited effect (Fig. 3B). We confirmed that the branch loss persisted in the D2 *daf-2* mutant, indicating that the branch loss is not due to the delayed development of mutant hermaphrodites (Fig. EV2C) (Mata-Cabana et al, 2022). The *daf-16; daf-2* double mutant exhibited a branching phenotype similar to the *daf-16* mutant alone,

confirming that *daf-16* acts downstream of *daf-2* in regulating PVP branch formation (Fig. 3B). Consistent with the role of insulin pathway in modulating PVP morphology, the *daf-16* mutant suppressed the wing-to-rod transition in adult hermaphrodites under dietary restriction (Fig. 3A). Since altering nutritional status at the adult stage did not lead to branch loss (Fig. EV2A), this suggests distinct roles of insulin signaling in controlling branch formation and branch morphology.

Next, we wanted to identify the functional site of DAF-16 in the PVP branching. *daf-16* has five isoforms: *daf-16a1/a2*, *daf-16b*, and *daf-16d/f* by alternative transcriptional start sites. While *daf-16a1/a2* and *daf-16d/f* isoforms are ubiquitously expressed and primarily associated with longevity (Kwon et al, 2010; Lin et al, 2001), *daf-16b* isoform is expressed in the pharynx, somatic gonad, and neurons for pharynx remodeling in dauers and neurite outgrowth (Christensen et al, 2011; Lee et al, 2001). Given its role in neurite development, we hypothesized that *daf-16b* is the primary isoform regulating PVP branch formation. Pan-neural expression of *daf-16b* partially restored the *daf-2* phenotype in *daf-16; daf-2* double mutants, while driving *daf-16b* expression specifically in PVP neurons recapitulated the *daf-2* mutant phenotype (Fig. 3C) (Christensen et al, 2011). This effect was specific to the *daf-16b* isoform, because expression of *daf-16f* in PVP neurons failed to suppress branching in the *daf-16; daf-2* double mutant (Fig. 3C). These results indicate that *daf-16b* acts cell-

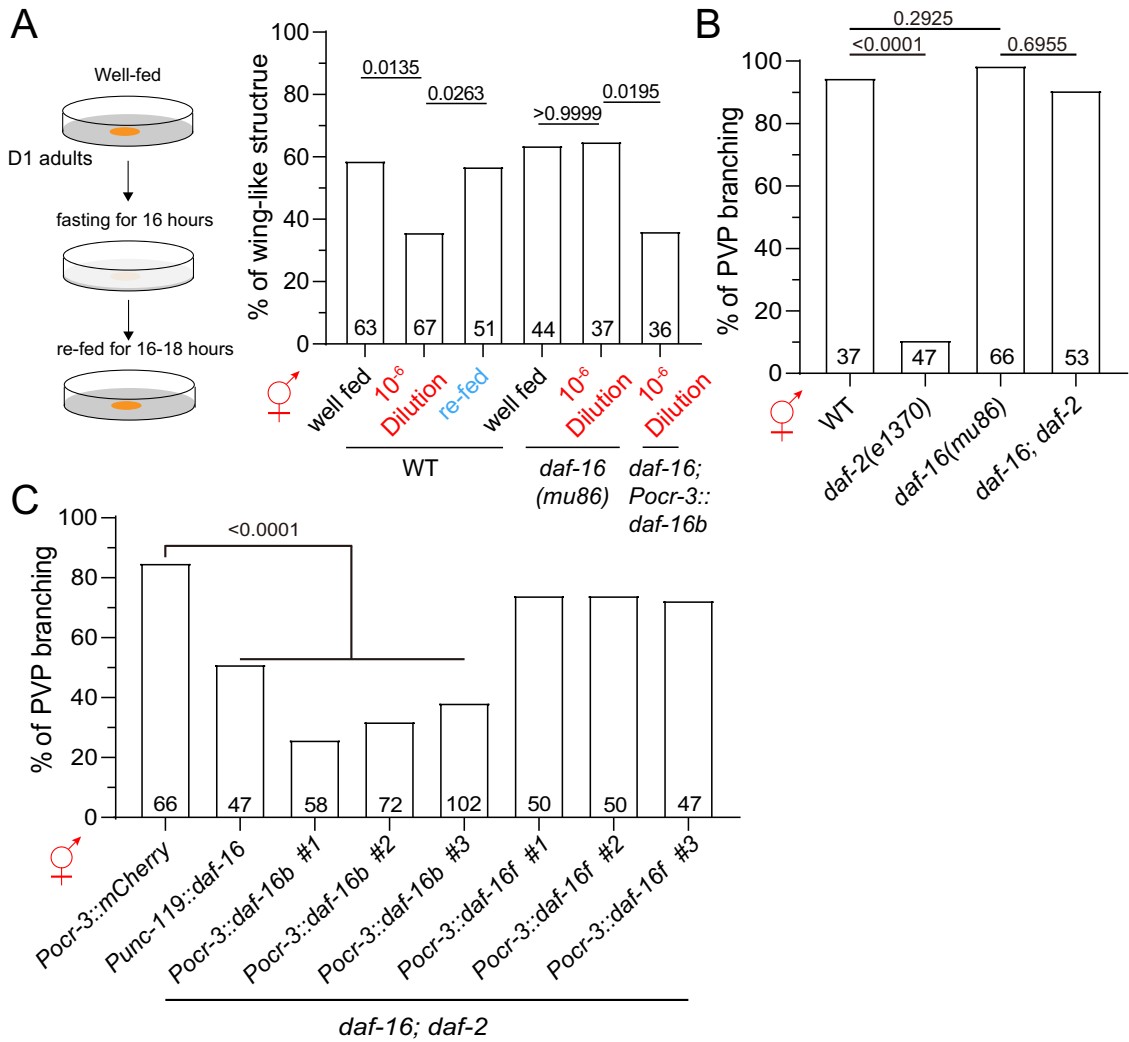

**Figure 3. Insulin pathway functions autonomously to regulate PVP branch formation and their morphology.**

(A) (left) A schematic diagram of fasting and re-feeding experiments. (right) Quantification of the percentage of wing-like PVP branches under nutritional conditions. Fisher's exact test. N number and P value are indicated. (B, C) Quantification of PVP branching at the indicated genotypes. Fisher's exact test. N number and P value are indicated. Source data are available online for this figure.

autonomously within PVP neurons to inhibit branch formation. Beyond branch initiation, expression of *daf-16b* in PVP neurons also rescued the wing-to-rod transition in the *daf-16* mutant under dietary restriction (Fig. 3A), suggesting a cell-autonomous role for *daf-16b* in regulating morphological dynamics in response to nutritional cues.

## Hermaphroditic fate of PVP neurons is necessary but not sufficient for branch formation

In *C. elegans*, biological sex influences the insulin signaling pathway mediated by *daf-2* and *daf-16*, which regulate the expression of chemoreceptors and drive sexually dimorphic behaviors (Wexler et al, 2020). *tra-1* encodes a master transcription factor to specify hermaphroditic fate triggered by the transmembrane protein TRA-2 (Wang and Kimble, 2001). In males, TRA-1 is targeted by the E3-ubiquitin ligase FEM-3 for proteasome degradation. To investigate the role of biological sex in PVP branch formation, we manipulated

the sex identity of PVP neurons by expressing FEM-3 or the intracellular domain of TRA-2 to selectively masculinize or feminize neurons (Fig. 4A). Such manipulations will not change the sexual identity of surrounding tissues. In hermaphrodites, masculinization of PVP neurons (PVP[masc]) significantly reduced TRA-1 expression and suppressed PVP branching (Fig. 4B,C), indicating that the hermaphroditic fate of PVP neurons is required for branch formation. Strikingly, branch loss in PVP[masc] hermaphrodites was restored by the *daf-16* mutation, indicating that *daf-16* functions downstream of sexual identity to control PVP branching (Fig. 4C). In addition, PVP[masc] hermaphrodites rarely formed wing-like branches even in the well-fed condition, suggesting that masculinization limits the nutritional plasticity of PVP morphology (Fig. EV2D). Conversely, feminization of PVP neurons (PVP[fem]) in males restored TRA-1 activity but did not promote branch formation even with the *daf-16* mutations (Fig. 4B,D). These results demonstrate that while the hermaphroditic fate of PVP

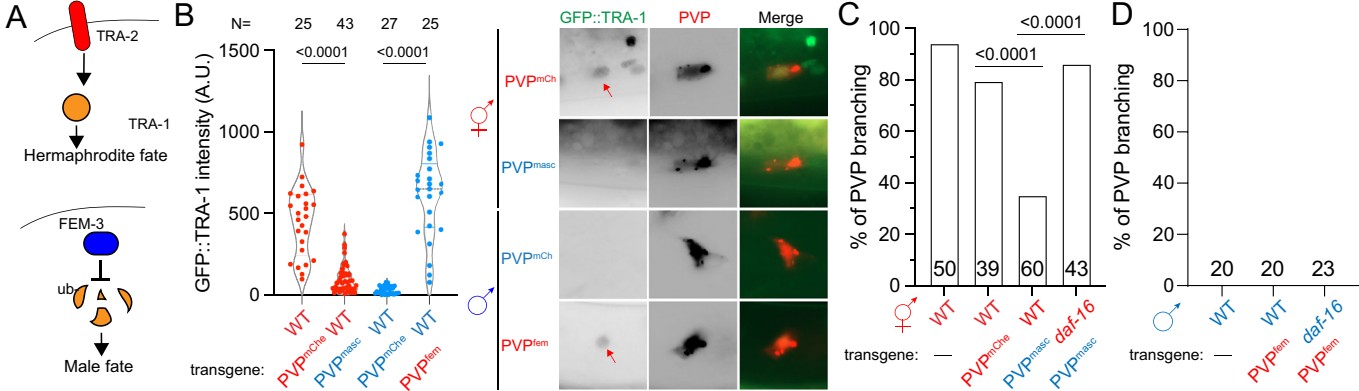

**Figure 4. Hermaphroditic fate of PVP neurons is necessary but not sufficient for PVP branch formation.**

(A) A schematic diagram of sexual transformation by overexpressing FEM-3 and TRA-2(IC) in *chcEx027[Pocr-3::FEM-3::SL2::mCherry]* and *chcEx009[Pocr-3::TRA-2::SL2::mCherry]*, respectively. (B) (left) Quantification of TRA-1 signal *via tra-1(ez72 [biotag::GFP::TEV::3xFlag::TRA-1])* in indicated genotypes. One-way ANOVA with Tukey correction. *N* number and *P* value are indicated. (right) Epifluorescent images of GFP::TRA-1 in males and hermaphrodites. Arrows indicate the TRA-1 signal in the PVP nucleus. Scale bar = 10 μm. (C, D) Quantification of PVP branching at indicated genotypes. Fisher's exact test. *N* number and *P* value are indicated. *N* indicates the number of biological repeats in this figure. Source data are available online for this figure.

neurons is necessary for branch formation and plasticity, it is not sufficient for PVP branching.

## uv1 Neuroendocrine cells, HSN neurons, and sex muscles are not required for PVP branching

Given that the feminization of PVP neurons in males is insufficient to induce PVP branching, we hypothesized that sex-specific tissues or cells near the vulva region, where PVP branches form, might contribute to branch development. PVP branches are positioned close to several cell types, including hermaphrodite-specific motor neurons (HSN), uv1 neuroendocrine cells, sex muscles, and vulval epithelial cells (Fig. 5A). We thus analyzed their contributions in hermaphrodites to explore their roles in promoting PVP branch formation.

Genetic ablation of HSN neurons by triggering apoptosis in the *egl-1* mutants caused only minor defects in PVP branch formation (Fig. 5B). Similarly, PVP branches remained intact in *pnc-1* mutants, where uv1 cells undergo necrosis due to accumulated nicotinamides (Fig. 5B) (Vrablik et al, 2009). Furthermore, sex muscles were found to be dispensable for PVP branching, as *egl-15* and *egl-17* mutants, which are defective in fibroblast growth factor (FGF) signaling for sex muscle migration, did not impair branch formation in hermaphrodites (Fig. 5B). These findings show that HSN neurons, neuroendocrine cells, and sex muscles near the vulva region do not play significant roles in promoting PVP branching.

## Vulva epithelia are essential and sufficient for PVP branch formation

Since HSN neurons, neuroendocrine cells, and vulval muscles were not involved in PVP branching, we investigated whether vulval epithelial cells play a critical role. Vulva formation involves extensive morphological changes in vulval epithelial cells, regulated by the interplay between MAPK and Notch signaling pathways (Sternberg, 2005). During early larval stages, precursor cells P3.p to P8.p are established and differentiate into three distinct epithelial

cell fates. P6.p adopts the primary cell fate, differentiating into VulE and VulF cells, while P5.p and P7.p adopt the secondary cell fate, giving rise to VulA, VulB, VulC, and VulD cells. The remaining precursor cells (P3.p, P4.p, and P8.p) adopt the tertiary cell fate and fuse with the hypodermis. Dysregulation of these signaling pathways often results in vulva-less or multi-vulva phenotypes (Schindler and Sherwood, 2013).

We found that PVP branches were absent in the *lin-39* vulva-less mutant (Fig. 5B), confirming the requirement of vulval epithelial cells for PVP branching. Notably, PVP branches were located close to VulE and VulF cells (Fig. 5C), suggesting that primary vulval cells are essential. To further test this, we used the mito-miniSOG system, which generates singlet oxygen to induce targeted cell toxicity upon blue light activation (440 nm), to ablate vulval cells (Xu and Chisholm, 2016). mito-miniSOG was driven by the *daf-6* and *cdh-3* promoters to selectively ablate VulC-F or VulE/F cells, respectively. Blue light was applied to transgenic and non-transgenic animals (mock controls) for 1 h prior to PVP branch formation at the L4 larval stage. While mock treatment had little effect on branch formation, mito-miniSOG-expressing transgenic hermaphrodites exhibited a marked loss of PVP branches (Fig. 5D; Appendix Fig. S3). These data indicate an essential role of VulE and VulF in PVP branching (Fig. 5D).

To understand whether vulval epithelial cells provide instructive cues for PVP branching, we studied the *let-60(gf)* mutant with heightened MAPK signaling that develops multiple vulvas by additional primary vulval cells in hermaphrodites. In this mutant, ectopic PVP branches formed exclusively near the vulva or extra-vulva structures (Fig. 5E), demonstrating that vulval cues guide PVP branching. We next asked whether the de novo formation of vulva-like structures could induce PVP branching in males. Since multiple vulvas were not observed in *let-60(gf)* males, we examined *lin-12(gf)* mutants, where excessive Notch signaling leads to the formation of protrusive pseudo-vulva structures in males (Sternberg, 2005). In approximately 50% of *lin-12(gf)* males, PVP neurons formed rod-like branches near these pseudo-vulva

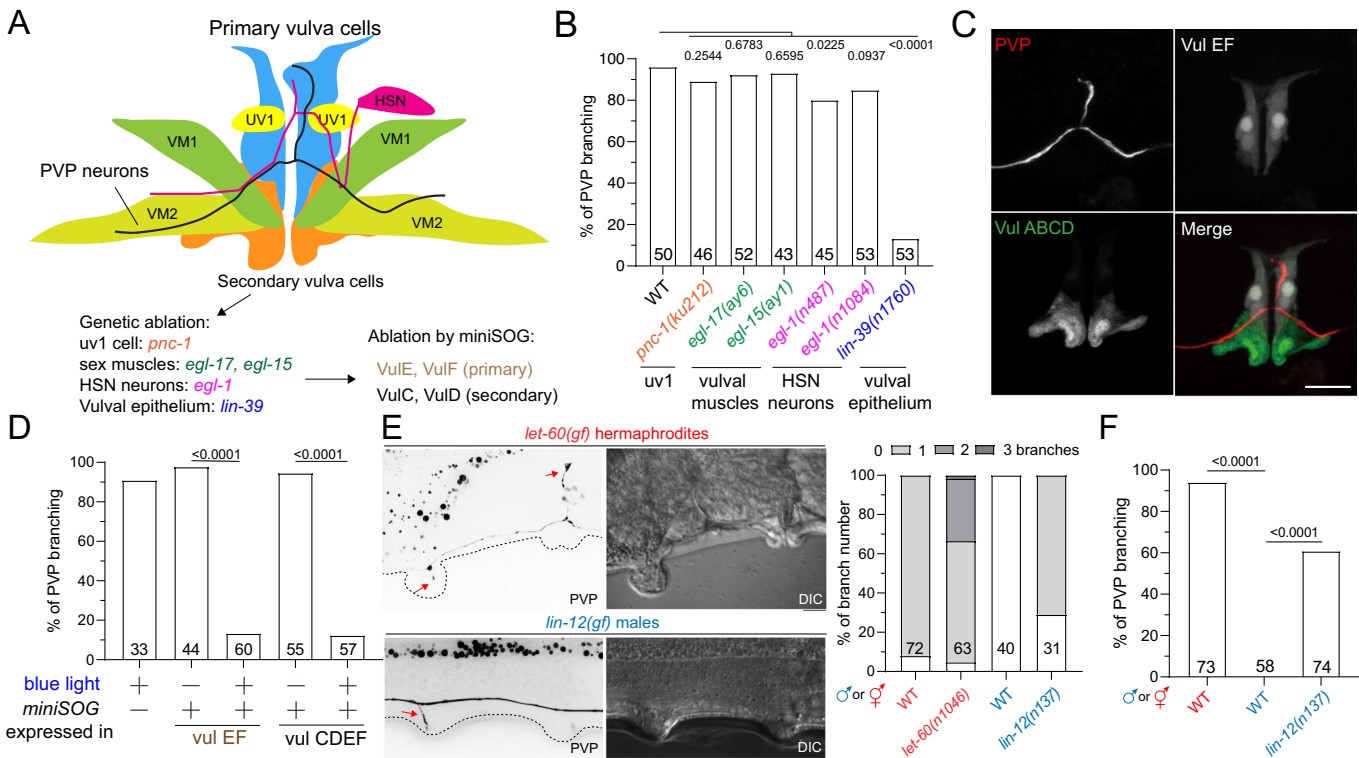

**Figure 5. Vulva epithelial cells instruct the PVP branch formation.**

(A) A schematic diagram of PVP and surrounding neurons, sex muscles, neuroendocrine cells, and vulva epithelial cells. (B) Quantification of PVP branching at the indicated genotypes. Fisher's exact test. *N* number and *P* value are indicated. (C) Z-projection of confocal fluorescent images of VulE, VulF, and PVP branches. Scale bar = 10 μm. (D) Quantification of PVP branching at indicated genotypes and experimental conditions. VulC, VulD, VulE, and VulF were ablated by the *chcEx162[Pdaf-6::mito::miniSOG::SL2::mKate]*, while VulE and VulF were ablated by the *chcEx160[Pcdh-3:: mito::miniSOG::SL2::mKate]*. Fisher's exact test. *N* number and *P* value are indicated. (E) (left) Z-projection epifluorescent images and DIC images of *let-60(gf)* hermaphrodites and *lin-12(gf)* males. Scale bar = 10 μm. Arrows indicate PVP branches. (right) Quantification of PVP branch number in indicated genotypes. Fisher's exact test. *N* number and *P* value are indicated. (F) Quantification of PVP branching at indicated genotypes and experimental conditions. Fisher's exact test. *N* number and *P* value are indicated. Biological replicates are present in this figure. Source data are available online for this figure.

regions, unlike disoriented protrusions typically observed in males (Fig. 5E,F). Together, these findings demonstrate that vulval epithelial cells, specifically VulE and VulF, provide critical instructive cues for PVP branching independent of the sexual identity of PVP neurons.

## Vulva cues and sex-regulated insulin pathway act in parallel but converge on F-actin assembly for PVP branching

How do vulval cues and the sex-regulated insulin pathway interact to shape sexually dimorphic PVP branching? To investigate this, we utilized a CRISPR-engineered, endogenously tagged transgenic strain from a previous study to measure TRA-1 expression (Bayer et al, 2020). We found that TRA-1 expression in PVP neurons gradually increased from the L2 to L4 stages and peaked in adulthood (Appendix Fig. S4). Disrupting vulva formation using a *lin-39* mutant or mito-miniSOG had no effect on TRA-1 expression (Fig. 6A). Increased vulva-like structures in the *let-60(gf)* mutant hermaphrodites also did not enhance TRA-1 expression. In addition, TRA-1 levels remained unchanged in *daf-2* mutants, consistent with the previous study that nutritional status has

minimal impact on the sexual state in the nervous system (Fig. 6A) (Bayer et al, 2020). These results suggest that vulval cues and the sex-regulated insulin pathway act in parallel to influence PVP morphogenesis.

Next, we analyzed F-actin enrichment in these mutants to further investigate the relationship between autonomous (*daf-2/daf-16* insulin signaling) and non-autonomous (*lin-39* vulval cues) factors in branch formation. F-actin was highly concentrated in wild-type animals at the future branching site during the L4 stage (Fig. 6B,C). However, this enrichment was absent in both *daf-2* and *lin-39* mutants, suggesting that autonomous and non-autonomous pathways converge on F-actin organization (Fig. 6B,C). Altogether, our study demonstrates that PVP branching is regulated by parallel autonomous and non-autonomous signaling pathways, highlighting a flexible mechanism for shaping sexually dimorphic neuronal morphogenesis.

## Loss of PVP branches modulates egg-laying behaviors in hermaphrodites

What is the physiological role of PVP branching? Given the localization of PVP branches within the vulval region, we

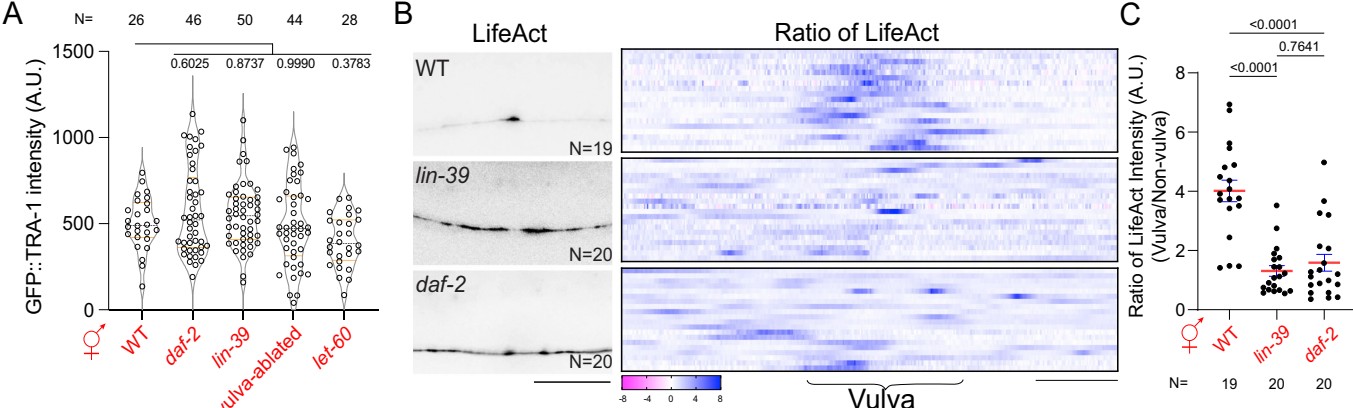

**Figure 6. Parallel insulin and epidermal cues pattern sexually dimorphic branching.**

(A) Quantification of GFP::TRA-1 signal *via tra-1(ez72 [biotag::GFP::TEV::3xFlag::TRA-1])* in indicated genotypes. One-way ANOVA with Tukey correction. *N* number and *P* value are indicated. (B) A heatmap showing the ratio of LifeAct in the PVP process in indicated genotypes. *N* number is indicated. Scale bar = 5 μm. Anterior is to the left. (C) Quantification of the ratio of LifeAct average intensity of the vulva and non-vulva region. One-way ANOVA with the Tukey correction. Error bar indicates SEM. *N* number and *P* value are indicated. *N* indicates the number of biological repeats in this figure. Source data are available online for this figure.

hypothesized that these structures are crucial for egg-laying behavior. To test this possibility, we measured egg-laying rates over a 2-h period. We found that hermaphrodites with masculinized PVP neurons, which lacked collateral branching, exhibited significantly reduced egg-laying rates, while expression of mCherry had minimal effect (Fig. 7A). The effect appears to be specific to egg-laying behavior, as fecundity was not substantially influenced by the sexual state of PVP neurons (Fig. EV3A). Notably, restoring PVP branching via *daf-16* mutations largely rescued the egg-laying defects caused by PVP masculinization (Fig. 7A). Together, these results indicate that PVP branch formation is essential for normal egg-laying behavior.

In *C. elegans*, serotonin released from HSN neurons initiates the active phase of egg laying by sensitizing vulval muscles to cholinergic input from VC neurons (Fig. 7B) (Garcia and Portman, 2016). Neuroendocrine uv1 cells modulate the egg-laying circuit by releasing tyramine, which suppresses HSN activity through the tyramine-gated chloride channel LGC-55, thereby terminating the active phase (Fig. 7B) (Yan et al, 2025). Strikingly, we found no physical contacts between PVP branches and HSN or VC neurons (Fig. 7C), suggesting that PVP branches modulate egg laying indirectly. Loss of PVP branches in PVP^masc hermaphrodites further reduced egg-laying rates in the *lgc-55* mutant (Fig. 7D), indicating that PVP branches act in parallel to the uv1-HSN axis. To probe their relationship with serotonin signaling, we exposed D1 hermaphrodites to exogenous serotonin in liquid buffer (Collins et al, 2016). As expected, while egg laying was attenuated in the absence of food, wild-type hermaphrodites still laid eggs when supplied with serotonin. By contrast, PVP^masc hermaphrodites lacking branches showed a reduced egg-laying rate even in the presence of serotonin (Fig. 7E), supporting that PVP branches act downstream or in parallel to HSN neurons. Collectively, these findings suggest that PVP branches contribute to the modulation of egg-laying circuitry downstream of serotonergic pathways critical for reproduction in hermaphrodites.

## Discussion

Our findings reveal that sexually dimorphic PVP branching was orchestrated independently by both autonomous insulin signaling and non-autonomous epidermal cues. Interestingly, we demonstrate that PVP branch morphology was highly plastic and responded to nutritional status in a *daf-16*-dependent manner. This suggests a dynamic link between metabolic state and neuronal structure, consistent with the broader role of insulin signaling in synaptic plasticity and neurite remodeling in other systems (Larrieu et al, 2014; Shimada-Niwa and Niwa, 2014). Insulin signaling and vulval cues converged on F-actin assembly, indicating that cytoskeletal remodeling is the integrative point for these parallel pathways. Lastly, we demonstrated that PVP branches are critical for normal egg-laying behaviors, highlighting the physiological importance of this sexually dimorphic neuronal structure. Altogether, our studies provide a paradigm where biological sex engages the insulin pathway for neuronal morphogenesis that facilitates the sex-specific behavior.

### The role of PVP neurons and their sexually dimorphic branches

In hermaphrodites, PVP neurons exhibit the highest degree of gap junction connectivity with DVC, PVT, and ciliated oxygen-sensing neurons AQR and PQR, which have openings in the coelomic cavity (Emmons, 2024). Although chemical synaptic inputs to PVP neurons are limited, these neurons primarily transmit signals to AVA, AVB, and PVC neurons to regulate locomotor activity (Emmons, 2024). Consistent with this, PVP neurons are implicated in modulating the transition between roaming and dwelling behaviors via the release of the neuropeptide PDF-1 (Flavell et al, 2013). While hermaphroditic PVP neurons show strong TRA-1 enrichment, electron microscopy has revealed only slight reductions in synaptic connections in males, with no apparent sexual

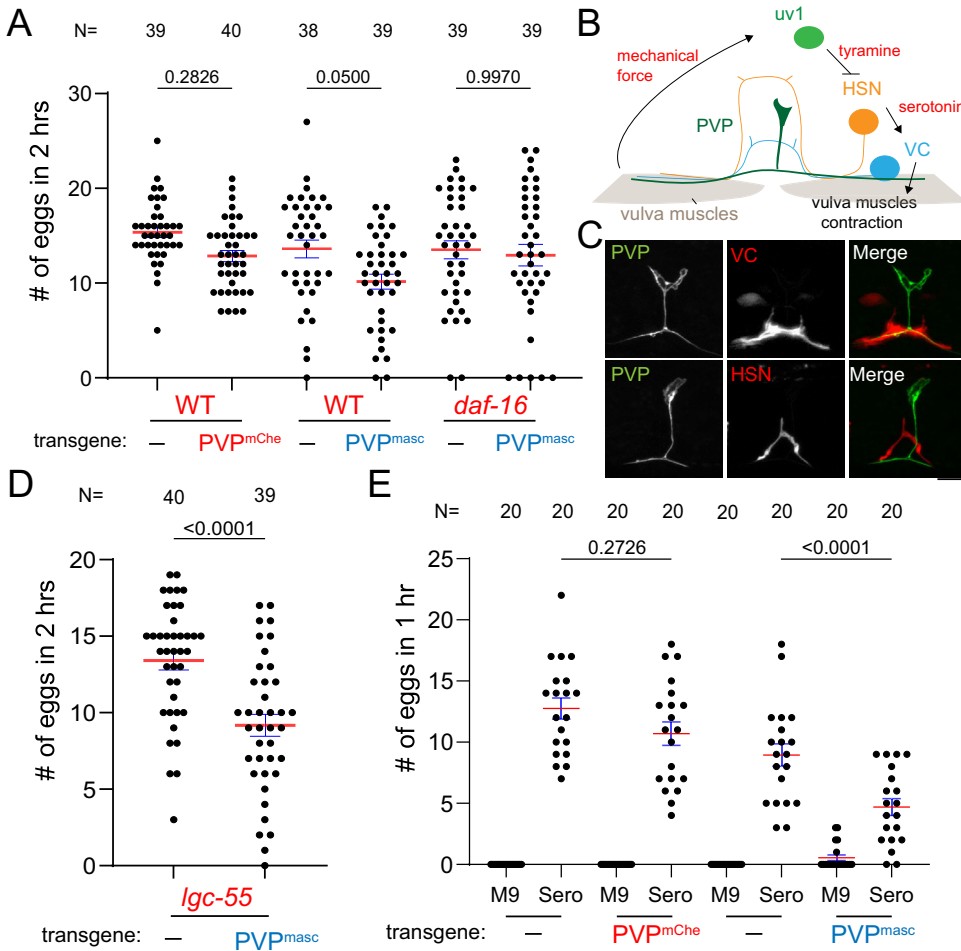

**Figure 7. PVP branches engage in egg-laying behavior.**

(A) Number of eggs laid with mCherry control carrying *chcEx299[Pocr-3::mCherry (7.5 ng); Pelt-2::NLS::BFP (50 ng)]*, masculinized PVP by overexpressing FEM-3 carrying *chcEx027[Pocr-3::FEM-3::SL2::mCherry]* and *daf-16; chcEx027[Pocr-3::FEM-3::SL2::mCherry]*. Dot represents one worm. Error bar indicates SEM. One-way ANOVA with Tukey correction. *N* number and *P* value are indicated. (B) A schematic diagram of the egg-laying circuit. (C) Z-projection epifluorescent images of VC/HSN colocalization. Scale bar = 10 μm. (D) Number of eggs laid with *lgc-55; chcEx027[Pocr-3::FEM-3::SL2::mCherry]*. Dot represents one worm. Error bar indicates SEM. Mann–Whitney *U*-test. *N* number and *P* value are indicated. (E) Liquid-based assay to quantify the number of eggs laid with *chcEx299[Pocr-3::mCherry(7.5 ng); Pelt-2::NLS::BFP(50 ng)]* and *chcEx027[Pocr-3::FEM-3::SL2::mCherry]* with or without 18.5 mM serotonin. Dot represents one worm. Error bar indicates SEM. One-way ANOVA with Tukey correction. *N* number and *P* value are indicated. *N* indicates the number of biological repeats in this figure. Source data are available online for this figure.

dimorphism in their overall synaptic architecture. However, our findings highlight the emergence of sexually dimorphic PVP branches, suggesting the potential for sexually dimorphic wiring contributing to distinct behavioral outputs.

Egg-laying behaviors are mainly controlled by HSN motor neurons and other surrounding VC neurons and uv1 neuroendocrine cells (Collins et al, 2016; Yan et al, 2025). Our data indicate that PVP branches modulate egg-laying behaviors presumably via peptidergic signaling. Interestingly, we did not observe physical contact between PVP branches, HSN, and VC neurons. This supports the view that extrasynaptic signaling is involved. While our results demonstrated the role of PVP branches in egg-laying circuits, the exact function of PVP branches remains unknown. Recently, mechanical sensation in VC, vulva muscles, and uv1 neuroendocrine cells serves as a feedback loop to coordinate egg-laying behaviors (Li and Chalfie, 1990; Medrano and Collins, 2023;

Yan et al, 2025). One tentative idea is that PVP branches also sense the mechanical force in this module during egg-laying to coordinate locomotion. However, more analyses are needed to substantiate this hypothesis in the future.

## Morphological plasticity of PVP neurons in response to physiological status

A striking observation in our study is the morphological plasticity of PVP branches, which exhibit two distinct structures: wing-like and rod-like. The wing-like morphology is more prominent under well-fed conditions, whereas the rod-like structure predominates in fasting hermaphrodites. This dynamic transition between the two morphologies may enable PVP neurons to modulate synaptic strength in response to nutritional status. Although the internal stimuli triggering this morphological plasticity remain unclear, our

data suggest these signals converge on the FOXO transcription factor DAF-16.

The influence of nutritional status on neuronal morphology is well-documented across species. In mammals, malnutrition involving omega-3 polyunsaturated fatty acids (PUFAs) reduces the complexity of apical dendrites in the prefrontal cortex (Larrieu et al, 2014). In *Drosophila*, the proper projection of serotonergic neurons, which regulate steroid hormone ecdysteroid production, depends on adequate nutrition (Shimada-Niwa and Niwa, 2014). In *C. elegans*, IL2 neurons form excessive dendritic branches during the dauer stage, mediated by the furin protein KPC-1 (Schroeder et al, 2013). Our findings add to this repertoire and offer a novel example of how nutritional and physiological cues influence neuronal structure. Further characterization of the mechanisms underlying this plasticity will deepen our understanding of how environmental and internal signals shape neuronal morphology and function.

## The interplay between biological sex and the insulin pathway

Our data suggest that biological sex regulates PVP branching through the *daf-16*/FOXO transcription factor, positioning the insulin pathway downstream of biological sex. Although our genetic and behavioral analyses support the view of sex-regulated insulin signaling, direct evidence of sexually dimorphic *daf-16* activity in PVP neurons remains absent. Notably, the expression of the food-sensing odor receptor *odr-10* is significantly enhanced upon starvation in both males and hermaphrodites (Wexler et al, 2020). However, the upregulation of *odr-10* exclusively depends on *daf-16* in males but not in hermaphrodites. Mechanistically, *daf-16* directly binds to the *odr-10* promoter to drive its transcription (Wexler et al, 2020). These findings support a model in which biological sex determines the sexually dimorphic response of insulin pathways, influencing neurodevelopment and behaviors. Sex-biased insulin responsiveness has also been observed in insects and mammals, contributing to sexually dimorphic traits such as body size and metabolism (Millington et al, 2021; Ortiz-Huidobro et al, 2021). This points to an evolutionarily conserved regulatory mechanism. Despite these genetic insights, further studies are needed to unravel the detailed mechanisms underlying the interplay between biological sex and the insulin pathway.

## Limitations of this study

Our phenotypic analyses revealed a transgenic toxicity associated with the *ocr-3* promoter (PVP-specific) that affected PVP branch morphogenesis, consistent with previous observations (Christie and Koelle, 2022). Although we minimized this effect by using a relatively low transgene concentration (7.5 ng/μL), a mild reduction in PVP branch formation and wing-to-rod transitions was still observed. Consequently, any manipulation of gene activity or expression of synaptic or cytoskeletal markers under the *ocr-3* promoter is likely to partially alter the normal morphology and function of PVP neurons. To address this limitation, future studies could employ CRISPR-mediated insertion of a single-copy transgene driven by the *ocr-3* promoter or identify alternative PVP-specific promoters with minimal toxicity for labeling and functional assays.

# Methods

### Reagents and tools table

| Reagent/resource | Reference or source | Identifier or catalog number |
|---|---|---|
| **Experimental models** | | |
| *E. coli*: Strain OP50 | Caenorhabditis Genetics Center (CGC) | WormBase: OP50 |
| *C. elegans*: Strain N2: wild isolate | Caenorhabditis Genetics Center (CGC) | WormBase: N2 |
| *C. elegans*: Strain NTU0021: chcSi1(Pocr-3::NeonGreen)/I | In this paper | |
| *C. elegans*: Strain NTU0188: chcEx104[Pocr-3::ida-1::gfp(7.5 ng); Ppdf-1::mKate(5 ng)] | In this paper | |
| *C. elegans*: Strain NTU0082: chcEx034(Pocr-3::gfp::cla-1::SL2::mKate(7.5 ng); Pelt-2::BFP(20 ng)) | In this paper | |
| *C. elegans*: Strain NTU0074: chcSi1(Pocr-3::NeonGreen)/I; chcEx032[Pocr-3::mCherry::RAB-3(7.5 ng) ; Pelt-2::BFP(50 ng)] | In this paper | |
| *C. elegans*: Strain NTU0461: chcSi1(Pocr-3::NeonGreen)/I; him-5(e1490)/V; chcEx251[Pocr-3::KAP-1::mCherry(7.5 ng); Pelt-2::BFP(50 ng)] | In this paper | |
| *C. elegans*: Strain NTU0447: chcEx248[Ppdf-1::mKate(5 ng); Pocr-3::gfp(7.5 ng)] | In this paper | |
| *C. elegans*: Strain NTU0135: chcEx072[Ppdf-1::mkate(5 ng); Pocr-3::mito::gfp(7.5 ng)] | In this paper | |
| *C. elegans*: Strain NTU0136: chcEx073[Ppdf-1::mkate(5 ng); Pocr-3::TGN-38::gfp(7.5 ng)] | In this paper | |
| *C. elegans*: Strain NTU0133: chcEx070[Ppdf-1::gfp(5 ng); Pocr-3::Lifeact::mkate(7.5 ng)] | In this paper | |
| *C. elegans*: Strain NTU0448: chcEx086[Pocr-3::gfp(50 ng);Pttx-3::rfp(35 ng)] | In this paper | |
| *C. elegans*: Strain NTU0108: chcEx055[Pocr-3::gfp(15 ng);Pttx-3::rfp(35 ng)] | In this paper | |
| *C. elegans*: Strain NTU0449: chcEx249[Pocr-3::gfp(7.5 ng);Pttx-3::rfp(30 ng)] | In this paper | |
| *C. elegans*: Strain NTU0114: chcSi1(Pocr-3::NeonGreen)/I; daf-2(e1370)/III | In this paper | |
| *C. elegans*: Strain NTU0106: chcSi1(Pocr-3::NeonGreen) daf-16(mu86)/I | In this paper | |
| *C. elegans*: Strain NTU0450: chcSi1(Pocr-3::NeonGreen) daf-16(mu86)/I; daf-2(e1370)/III | In this paper | |

| Reagent/resource | Reference or source | Identifier or catalog number |
|---|---|---|
| *C. elegans*: Strain NTU0306: *chcSi1(Pocr-3::NeonGreen) daf-16(mu86)/I; daf-2(e1370)/III; chcEx156[Pocr-3::mCherry(7.5 ng); Pttx-3::RFP(30 ng)]* | In this paper | |
| *C. elegans*: Strain NTU0176: *chcSi1(Pocr-3::NeonGreen) daf-16(mu86)/I; daf-2(e1370)/III; muEx169[unc-119p::GFP::daf-16 + rol-6(su1006)]* | In this paper | |
| *C. elegans*: Strain NTU0249: *chcSi1(Pocr-3::NeonGreen) daf-16(mu86)/I; daf-2(e1370)/III; chcEx130[Pocr-3::daf-16b(7.5 ng); Pelt-2::NLS::tagBFP(30 ng)#1]* | In this paper | |
| *C. elegans*: Strain NTU0250: *chcSi1(Pocr-3::NeonGreen) daf-16(mu86)/I; daf-2(e1370)/III; chcEx130[Pocr-3::daf-16b(7.5 ng); Pelt-2::NLS::tagBFP(30 ng)#2]* | In this paper | |
| *C. elegans*: Strain NTU0251: *chcSi1(Pocr-3::NeonGreen) daf-16(mu86)/I; daf-2(e1370)/III; chcEx130[Pocr-3::daf-16b(7.5 ng); Pelt-2::NLS::tagBFP(30 ng)#3]* | In this paper | |
| *C. elegans*: Strain NTU0517: *chcSi1(Pocr-3::NeonGreen) daf-16(mu86)/I; daf-2(e1370)/III; chcEx291[Pocr-3::daf-16f(7.5 ng); Pelt-2::NLS::tagBFP(30 ng)#1]* | In this paper | |
| *C. elegans*: Strain NTU0518: *chcSi1(Pocr-3::NeonGreen) daf-16(mu86)/I; daf-2(e1370)/III; chcEx292[Pocr-3::daf-16f(7.5 ng); Pelt-2::NLS::tagBFP(30 ng)#2]* | In this paper | |
| *C. elegans*: Strain NTU0519: *chcSi1(Pocr-3::NeonGreen) daf-16(mu86)/I; daf-2(e1370)/III; chcEx293[Pocr-3::daf-16f(7.5 ng); Pelt-2::NLS::tagBFP(30 ng)#3]* | In this paper | |
| *C. elegans*: Strain NTU0527: *chcSi1(Pocr-3::NeonGreen) daf-16(mu86)/I; chcEx130[Pocr-3::daf-16b(7.5 ng); Pelt-2::NLS::tagBFP(30 ng)#3]* | In this paper | |
| *C. elegans*: Strain NTU0323: *tra-1(ez72[biotag::GFP::TEV::3xflag::tra-1])/III; zuIs236; chcEx168[Pocr-3::mCherry(7.5 ng); Pttx-3::RFP(30 ng)]* | In this paper | |
| *C. elegans*: Strain NTU0454: *tra-1(ez72[biotag::GFP::TEV::3xflag::tra-1])/III; zuIs236; Pocr-3::fem-3::SL2::mCherry(7.5 ng); Pttx-3::RFP(50 ng)* | In this paper | |
| *C. elegans*: Strain NTU0455: *tra-1(ez72[biotag::GFP::TEV::3xflag::tra-1])/III; zuIs236; Pocr-3::tra-2::SL2::mCherry(7.5 ng); Pttx-3::RFP(50 ng)* | In this paper | |
| *C. elegans*: Strain NTU0023: *chcEx009[Pocr-3::TRA-2::SL2::mCherry]* | In this paper | |

| Reagent/resource | Reference or source | Identifier or catalog number |
|---|---|---|
| *C. elegans*: Strain NTU0064: *chcEx027[Pocr-3::fem-3::SL2::mCherry(7.5 ng); Pttx-3::RFP(50 ng)]* | In this paper | |
| *C. elegans*: Strain NTU0522: *chcEx295[Pocr-3::fem-3::SL2::mkate(7.5 ng); Pelt-2::NLS::tagBFP(50 ng)#1]* | In this paper | |
| *C. elegans*: Strain NTU0523: *chcEx296[Pocr-3::fem-3::SL2::mkate(7.5 ng); Pelt-2::NLS::tagBFP(50 ng)#2]* | In this paper | |
| *C. elegans*: Strain NTU0453: *daf-16(mu86)/I; chcEx027[Pocr-3::fem-3::SL2::mCherry(7.5 ng); Pttx-3::RFP(50 ng)]* | In this paper | |
| *C. elegans*: Strain NTU0521: *daf-16(mu86)/I; chcEx009[Pocr-3::TRA-2::SL2::mCherry]* | In this paper | |
| *C. elegans*: Strain NTU0206: *chcSi1(Pocr-3::NeonGreen)/I; pnc-1(ku212)/IV* | In this paper | |
| *C. elegans*: Strain NTU0144: *chcSi1(Pocr-3::NeonGreen)/I; egl-17(ay6)/X* | In this paper | |
| *C. elegans*: Strain NTU0211: *chcSi1(Pocr-3::NeonGreen)/I; egl-15(ay1)/X* | In this paper | |
| *C. elegans*: Strain NTU0115: *chcSi1(Pocr-3::NeonGreen)/I; egl-1(n487)/V* | In this paper | |
| *C. elegans*: Strain NTU0452: *chcSi1(Pocr-3::NeonGreen)/I; egl-1(n1084)/V* | In this paper | |
| *C. elegans*: Strain NTU0113: *chcSi1(Pocr-3::NeonGreen)/I; lin-39(n1760)/III* | In this paper | |
| *C. elegans*: Strain NTU0321: *mfIs4[egl-17::YFP + daf-6::CFP + unc-119(+)]; chcEx166[Pocr-3::mCherry(7.5 ng); Pttx-3::RFP(30 ng)]* | In this paper | |
| *C. elegans*: Strain NTU0315: *chcSi1(Pocr-3::NeonGreen)/I; chcEx162[Pdaf-6::miniSOG::SL2::mKate(30 ng); Pttx-3::RFP(30 ng)]* | In this paper | |
| *C. elegans*: Strain NTU0313: *chcSi1(Pocr-3::NeonGreen)/I; chcEx160[Pcdh-3::miniSOG::SL2::mKate(30 ng); Pttx-3::RFP(30 ng)]* | In this paper | |
| *C. elegans*: Strain NTU0175: *chcSi1(Pocr-3::NeonGreen)/I; let-60(n1046)/IV* | In this paper | |
| *C. elegans*: Strain NTU0173: *chcSi1(Pocr-3::NeonGreen)/I; lin-12(n137)/III; him-5(e1490)/V* | In this paper | |
| *C. elegans*: Strain NTU0456: *tra-1(ez72[biotag::GFP::TEV::3xflag::tra-1]) daf-2(e1370)/III; zuIs236; chcEx168[Pocr-3::mCherry(7.5 ng); Pttx-3::RFP(30 ng)]* | In this paper | |

| Reagent/resource | Reference or source | Identifier or catalog number |
|---|---|---|
| *C. elegans*: Strain NTU0457: *tra-1(ez72[biotag::GFP::TEV::3xflag::tra-1]) lin-39(n1760)/III; zuIs236; chcEx168[Pocr-3::mCherry(7.5 ng); Pttx-3::RFP(30 ng)]* | In this paper | |
| *C. elegans*: Strain NTU0463: *tra-1(ez72[biotag::GFP::TEV::3xflag::tra-1])/III; zuIs236; chcEx253[Pdaf-6::miniSOG::SL2::mKate(30 ng); Pttx-3::RFP(30 ng)]* | In this paper | |
| *C. elegans*: Strain NTU0459: *tra-1(ez72[biotag::GFP::TEV::3xflag::tra-1])/III; let-60(n1046)/IV; zuIs236; chcEx168[Pocr-3::mCherry(7.5 ng); Pttx-3::RFP(30 ng)]* | In this paper | |
| *C. elegans*: Strain NTU0332: *lin-39(n1760)/III; chcEx070[Ppdf-1::gfp(5 ng); Pocr-3::Lifeact::mkate(7.5 ng)]* | In this paper | |
| *C. elegans*: Strain NTU0259: *daf-2(e1370)/III; chcEx070[Ppdf-1::gfp(5 ng); Pocr-3::Lifeact::mkate(7.5 ng)]* | In this paper | |
| *C. elegans*: Strain NTU0524: *lgc-55(tm2913)/V; chcEx027[Pocr-3::fem-3::SL2::mCherry(7.5 ng); Pttx-3::RFP(50 ng)]* | In this paper | |
| *C. elegans*: Strain NTU0525: *chcEx299[Pocr-3::mCherry(7.5 ng); Pelt-2::NLS::BFP(50 ng)]* | In this paper | |
| *C. elegans*: Strain NTU0142: *chcSi1(Pocr-3::NeonGreen)/I; chcEx079[Pnlp-3::mCherry(50 ng); Pelt-2::NLS::BFP(5 ng)]* | In this paper | |
| *C. elegans*: Strain NTU0526: *chcSi1(Pocr-3::NeonGreen)/I; chcEx300[Pcat-1::mKate(40 ng); Pelt-2::NLS::BFP(50 ng)]* | In this paper | |
| **Oligonucleotides and other sequence-based reagents** | | |
| PCR primers | In this paper | Table EV1 |
| **Chemicals, enzymes and other reagents** | | |
| Serotonin hydrochloride | Sigma | Cat#9523 |
| **Software** | | |
| GraphPad Prism | 10.4.1 | |
| ZEN 3.6 | Zeiss | |
| ImageJ | https://imagej.net/ij/ | |
| **Other** | | |
| LSM 780 Confocal microscopy | Zeiss | |
| Zeiss Imager M2 microscope | Zeiss | |
| Axiocam 705 Mono camera | Zeiss | |
| Apoptome 3 | Zeiss | |

## Methods and protocols

### Maintenance of *C. elegans* strains

*C. elegans* strains were cultured and maintained as previously described (Brenner, 1974). In brief, animals were cultivated on nematode growth medium (NGM) plates seeded with *E. coli* OP50 bacteria at 20 °C. A table of mutant alleles and transgenic strains used in this study is available in the reagents and tools table. In order to control the nutritional status of worms, L4 hermaphrodites were cultivated on OP50 ($OD_{600} = 0.6$) diluted by LB medium (1:100 or 1:1000) for 16–18 h. Re-feeding experiments were conducted by transferring starved animals to the plate with non-diluted OP50 ($OD_{600} = 0.6$) for 16–18 h.

### Germline transformation

Germline transformation was performed by microinjecting plasmids into the gonads as described (Mello et al, 1991). In brief, the gonads of D1 adult worms were visualized by an inverted microscope (Zeiss, Axio Observer) at 40X objective, and the DNA mixture was injected through a microinjector (Eppendorf FemtoJet 4X). Isolated progeny carrying transgenes for more than two generations were considered independent transgenic lines for experiments.

### Molecular cloning

*ocr-3* (0.5 kb), *pdf-1* (3 kb), *unc-119* (1.6 kb), *daf-6* (3.5 kb), *cdh-3* (0.5 kb), *cat-1 (3 kb)*, and *nlp-3 (3.5 kb)* promoters were cloned into the pPD95.75 vector to label PVP, *pdf-1* expressing neurons, pan-neurons, vulE/vulF, and VulC/VulD/VulE/VulF, respectively (Barrios et al, 2012; Kirouac and Sternberg, 2003; Lorenzo et al, 2020). cDNA of CLA-1, RAB-3, DAF-16b, and DAF-16f was amplified from the cDNA library prepared by SuperScript III First-Strand Synthesis System (Thermo Fisher, Catalog number: 18080051) and cloned into pPD95.75-based plasmids with specific promoters. Primers used were listed in Table EV1.

### Single copy insertion by transgenic CRISPR

The single copy insertion of *Pocr-3::NeonGreen* was generated via a modified method derived from single-copy knock-in loci for defined gene expression (SKI LODGE) (Silva-Garcia et al, 2019). Specifically, *Pocr-3::NeonGreen* was cloned into a plasmid carrying homologous arms of the safe harbor on chromosome I (at cM: -5.31) and a hygromycin-resistant gene. This plasmid was coinjected with *Peft-3::Cas9::sgRNA* (gRNA: GGAAATCGCC-GACTTGCGAGG) and coinjection marker *Pblmp-1::GFP*. F1 progeny after microinjection were cultivated at 20 °C until starvation, and starved worms were transferred to fresh NGM plates with 0.3 mg/ml hygromycin to select non-GFP viable animals. The successful insertion was then validated by PCR genotyping. Primers are listed below: seqF (5'-tttgggaatgaagttgttta-gaaacg-3') and seqR (5'-tttgggaatgaagttgtttagaaacg-3').

### Image acquisition

Confocal fluorescent images were acquired by LSM 780 Confocal microscopy (LSM 780, Zeiss). Epifluorescent images were acquired by a Zeiss Imager M2 microscope with an Axiocam 705 Mono camera. Z-stack epifluorescent images were acquired by a Zeiss Imager M2 microscope equipped with an Apotome 3 system. Images were analyzed using ZEN Blue 3.6 software (Zeiss).

### Masculinization and feminization of PVP neurons

PVP neuron masculinization was induced by overexpressing *fem-3*, which encodes an E3-ubiquitin ligase that targets the transcription factor TRA-1 for degradation, using the transgene *chcEx027[Pocr-3::FEM-3::SL2::mCherry]*. Feminization was achieved by overexpressing the intracellular cytosolic fragment (IC) of *tra-2* using the transgene *chcEx009[Pocr-3::TRA-2::SL2::mCherry]*.

### Genetic ablation by mito-miniSOG

Synchronized L4 larvae carrying *Pcdh-3::miniSOG::SL2::mKate* or *Pdaf-6::miniSOG::SL2::mKate* were collected and transferred to 1 mL M9 solution on an empty NGM plate. Then, worms in solution were treated with blue light (440 nm; ~80 mW/m$^{-2}$) for an hour. Animals were recovered on solid plates, and their PVP branching phenotype was scored. Non-transgenic animals were treated with the same conditions as a mock control.

### Quantification of PVP branching phenotypes

Quantification of PVP branching.  About 30–50 D1 adult animals were quantified by picking L4 larvae a day before scoring. Animals were then mounted on a 5% agarose pad and paralyzed with 1% sodium azide. Images were taken by ZEN Blue 3.6 software using a 63X objective in a Zeiss Imager M2 microscope with an Axiocam 705 Mono camera. We defined a mature branch as one that has a protrusion longer than 8 μm. The percentage of PVP branching in a population was subsequently scored.

Semi-quantification of PVP branch position.  Adult animals were segmented into five regions of equal length. The percentage of PVP branches in each bin was scored.

Quantification of PVP branch morphology.  The width of branches defined the semi-quantification of wing- or rod-like structure. A branch width longer than 2 μm was scored as a wing-like structure, and a branch width shorter than 2 μm was counted as a rod-like structure. The percentage of PVP branching in a population was subsequently scored. To quantify F-actin distribution in wing- and rod-like branches, we measured the ratio of the length occupied by F-actin and diffusible GFP from the tip to the base of PVP branches.

Quantification of fluorescent signals.  Measurement of the average fluorescent intensity of the LifeAct signal was conducted by ImageJ. The vulva slit or P6.p vulva precursor cells were used as a landmark and were set as a midpoint. Then, the fluorescent signals were obtained using the line scan function in ImageJ. The measurement covered the region spanning 15 μm in length anterior or posterior to the midpoint. To quantify the enrichment of F-actin signal, the average of fluorescent signals within the vulva region (5 μm) was compared with the rest (25 μm). The average fluorescent intensity of GFP::TRA-1 signal was measured by Zen from the indicated cells by labeling the nucleus as ROI.

### Egg-laying behavioral assays

At least 20 L4 hermaphrodites were staged 1 day prior to the assay. For plate-based assays, a single hermaphrodite was transferred to an OP50-seeded NGM plate (18 μL bacterial lawn), and the number of eggs laid was counted after 2 h. For liquid-based assays, 100 μL of M9 buffer was used as a control, and 100 μL of pH 7.0, 18.5 mM serotonin hydrochloride soluble in water (Sigma-Aldrich, CAS Number #153-98-0) was added to each well of a 96-well plate as described previously (Collins et al, 2016). Each well contained a single hermaphrodite, and eggs were counted after 1 h of incubation.

### Statistical analysis

One-way ANOVA with Tukey correction, Student *t*-test, Mann–Whitney *U*-test, and Fisher's exact test were conducted by Prism (Version 10.4.1) as indicated in Figure legends.

### Declaration of generative AI and AI-assisted technologies in the writing process

During the preparation of this work, the author(s) used ChatGPT and Grammarly in order to improve the readability of this manuscript. After using this tool/service, the authors reviewed and edited the content as needed and take full responsibility for the content of the publication.

## Data availability

All data were included in the article and/or supplementary information. All raw data were deposited to Figshare (https://doi.org/10.6084/m9.figshare.28479827).

The source data of this paper are collected in the following database record: biostudies:S-SCDT-10_1038-S44319-025-00608-0.

## Peer review information

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

## Acknowledgements

We thank members of the Chen lab for their insightful discussion. We also thank Han Wang, Shih-Peng Chan, Yi-Chun Wu, and Chun-Liang Pan for providing reagents. Some strains were provided by the *C. elegans* Genetics Center, funded by the NIH Office of Research Infrastructure Programs (P40 OD010440). We are grateful for the technical assistance of Technology Commons, the College of Life Science, and the National Taiwan University with confocal microscopy (CLSM). We thank the *C. elegans* Core Facility of the National Core Facility for Biopharmaceuticals, Ministry of Science and Technology in Taiwan. This study was funded by the Ministry of Science and Technology, Taiwan, to C-HC (NSTC 112-2636-B-002-007, NSTC 113-2636-B-002-003, and NSTC 114-2311-B-002-024).

## Author contributions

**Jia-Bin Yang**: Conceptualization; Formal analysis; Investigation; Methodology; Writing—original draft; Writing—review and editing. **Rui-Tsung Chen**: Conceptualization; Investigation. **Yun-Yu Chen**: Validation; Investigation. **Yun-Hsien Lin**: Validation. **Chun-Hao Chen**: Conceptualization; Formal analysis; Supervision; Funding acquisition; Investigation; Writing—original draft; Writing—review and editing.

Source data underlying figure panels in this paper may have individual authorship assigned. Where available, figure panel/source data authorship is listed in the following database record: biostudies:S-SCDT-10_1038-S44319-025-00608-0.

## Disclosure and competing interests statement

The authors declare no competing interests.

# Expanded View Figures

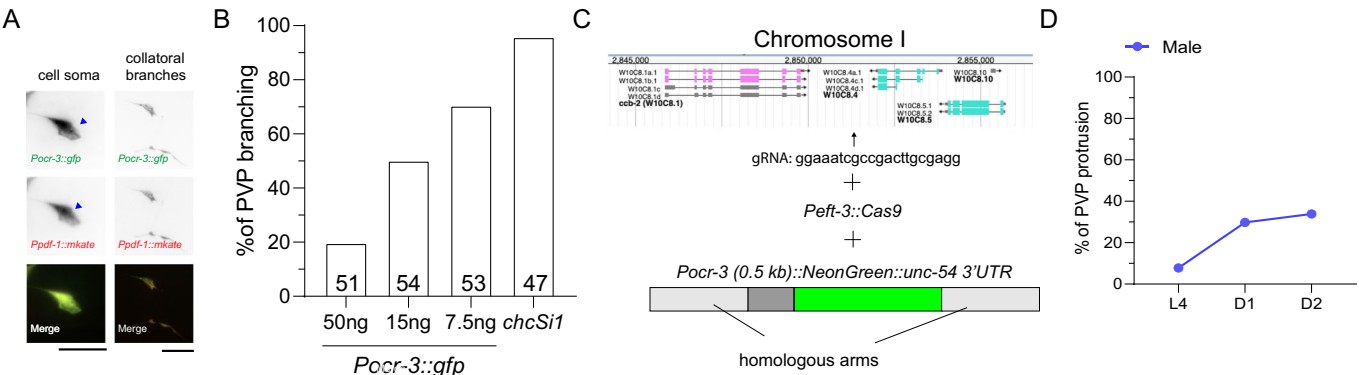

**Figure EV1. Characterization of transgenic strains with PVP-specific fluorescent reporters.**

(A) Z-projection of confocal fluorescent images of PVP neurons and markers for *pdf-1* neuropeptide in the *chcEx248[Ppdf-1::mKate; Pocr-3::gfp]*. Arrows indicate PVP branches. Scale bar = 10 μm. (B) Quantification of PVP branching with indicated transgenes. Fisher's exact test. N number and P value are indicated. (C) A schematic diagram for CRISPR-engineered single copy insertion of *Pocr-3::NeonGreen* on chromosome I. (D) Quantification of sexually dimorphic PVP branching at the indicated developmental stages. N numbers are indicated. N indicates the number of biological repeats in this figure. Source data are available online for this figure.

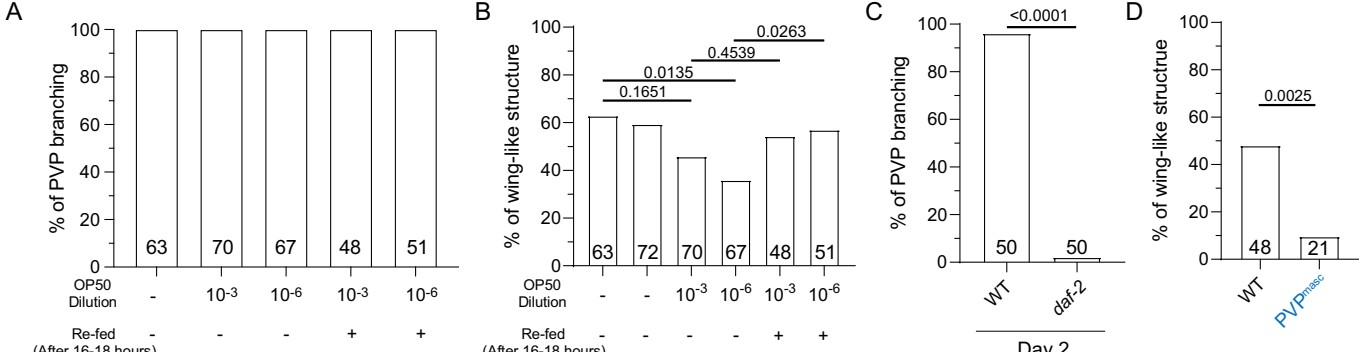

**Figure EV2. Nutritional status influences PVP branch morphology.**

(A) Quantification of wing-like branches at different nutritional statuses. (B) Quantification of PVP branching at different nutritional statuses. Fisher's exact test. N number and P value are indicated. (C) Quantification of PVP branching of D2 adults in indicated genotypes. Fisher's exact test. N number and P values are indicated. (D) Quantification of wing-like PVP branching in indicated genotypes. Fisher's exact test. N number and P values are indicated. N indicates the number of biological repeats in this figure. Source data are available online for this figure.

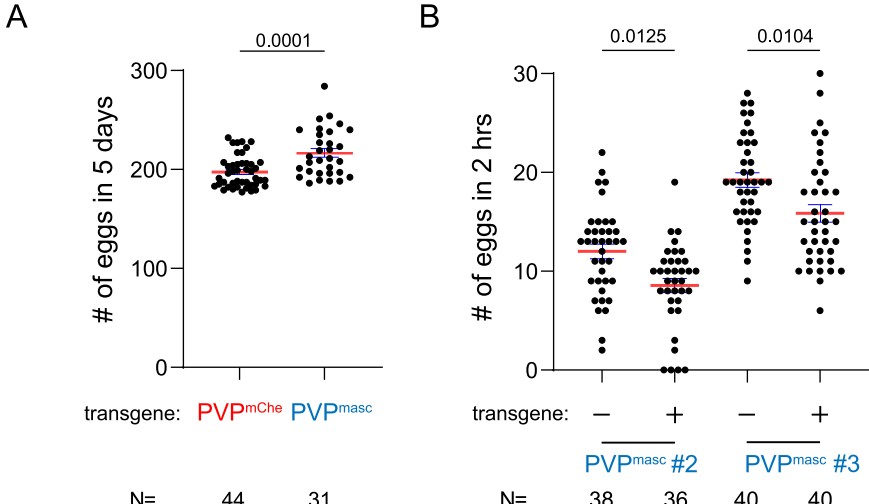

**Figure EV3. Egg-laying behaviors are modulated by PVP branches.**

(A) Quantification of total fertilized eggs with indicated genotypes. Dot represents one worm. Error bar indicates SEM. Mann–Whitney *U*-test. *N* number and *P* values are indicated. (B) Number of eggs laid with masculinized PVP by overexpressing FEM-3 carrying *chcEx295[Pocr-3::fem-3::SL2::mkate(7.5 ng); Pelt-2::NLS::tagBFP(50 ng)#1]* and *chcEx296[Pocr-3::fem-3::SL2::mkate(7.5 ng); Pelt-2::NLS::tagBFP(50 ng)#2]*. Error bar indicates SEM. *N* number and *P* values are indicated. One-way ANOVA with Tukey correction. Biological replicates are present in this figure. *N* indicates the number of biological repeats in this figure. Source data are available online for this figure.

