## [Peer Review File · EMBO Reports]

Insulin and Epidermal Signals Independently Shape Sexually Dimorphic Neurite Branching in *C. elegans*

Jia-Bin Yang, Rui-Tsung Chen, Yun-Yu Chen, Yun-Hsien Lin, and Chun-Hao Chen

Corresponding author(s): Chun-Hao Chen (chunhaochen@ntu.edu.tw)

Review Timeline:

Submission Date:	21st Mar 25
Editorial Decision:	28th Mar 25
Appeal Received:	31st Mar 25
Editorial Decision:	23rd Apr 25
Revision Received:	26th Aug 25
Editorial Decision:	1st Oct 25
Revision Received:	3rd Oct 25
Accepted:	13th Oct 25

Editor: Esther Schnapp

Transaction Report:

27th Mar 2025

Dear Dr. Chen,

Thank you for the submission of your manuscript to EMBO reports. I have now read and discussed it with my colleagues here, and I am sorry to say that we all agree that it is not well suited for us.

We note that your study investigates sexually dimorphic collateral branching of PVP cholinergic interneurons in *Caenorhabditis elegans*. You show that PVP branching is largely absent in males, that PVP branches have both axonal and dendritic properties, that PVP branch morphology adapts to nutritional status, that *daf-2* promotes PVP branching, that hermaphrodite identity of PVP neurons is required but not sufficient for branch formation, that the vulval cells VulE and VulF are necessary for branching, and that both insulin signaling and vulval cues converge on F-actin cytoskeletal remodeling. We recognize that this will be of interest to researchers in the field.

However, we also note that it remains unclear what the physiological relevance of the PVP branches is, and how it relates to sexual identity. If data on how the sexually dimorphic branches regulate for example worm locomotion or egg laying could be provided, we would be happy to send your ms for in-depth review. As it stands, we think that the ms would be better suited for our sister journal Life Science Alliance.

Life Science Alliance (<http://www.life-science-alliance.org/>) is our broad scope Open Access journal published in partnership between the EMBO-, Rockefeller University-, and Cold Spring Harbor Laboratory Presses. Tim Fessenden, Executive Editor of Life Science Alliance (t.fessenden@life-science-alliance.org) would be pleased to send your manuscript for in-depth peer review; no reformatting is required. We very much hope that you will be interested in this option: please follow the link below for transfer.

For EMBO reports, I am sorry that I cannot be more positive this time, and I thank you once more for your interest in our journal.

Yours sincerely,

** As a service to authors, EMBO Press provides authors with the ability to transfer a manuscript that one journal cannot offer to publish to another journal, without the author having to upload the manuscript data again. To transfer your manuscript to another EMBO Press journal using this service, please click on Link Not Available

Dear Dr. Esther Schnapp,

We are pleased to re-submit our manuscript (original submission number: EMBOR-2025-61581), "**Insulin and Epidermal Signals Independently Shape Sexually Dimorphic Neurite Branching in *C. elegans***" for consideration in *EMBO Reports* with additional behavioral analysis suggested from our prior communication. We have integrated the behavioral data and modified our manuscript accordingly.

Sexually dimorphic neuronal structures arise through complex interactions between genetic programs and environmental cues, yet the molecular mechanisms governing their development remain elusive. Using *C. elegans* as a genetically tractable model, we uncover parallel and independent roles of autonomous insulin signaling and non-autonomous epidermal cues in regulating sexually dimorphic axonal branching.

We show that hermaphrodite-specific PVP interneuron branching exhibits both axonal and dendritic properties, selectively accumulating dense core vesicles for neuropeptide signaling. PVP branching requires intrinsic sex identity via the DAF-2/DAF-16 insulin pathway, while nutritional states dynamically reshape mature PVP morphology, revealing metabolic control over neuronal plasticity. Independently, epithelial cues from vulval cells induce PVP branching even in males, overriding intrinsic sexual identity. Both mechanisms converge on F-actin assembly, suggesting a conserved cytoskeletal framework for sex-specific neural development. Lastly, we demonstrate that sexually dimorphic branching engages in egg-laying behaviors in hermaphrodites, bridging the neurite structure to a sex-specific reproductive behavior.

Our study thus provides a detailed molecular framework linking sexual dimorphism, neuronal plasticity, internal states, and behaviors. These findings offer new insights into how hormonal and environmental signals interact to shape sexually dimorphic neural circuits, with broader implications for understanding metabolic regulation of neuronal morphology across species.

Given the broad implications of our work for understanding neural plasticity and sexual dimorphism, we believe our study aligns with the high standards and interdisciplinary readership of *EMBO Reports*. We appreciate your time and consideration and look forward to hearing you soon.

Best Regards,

Jia-Bin Yang and Chun-Hao Chen

Institute of Cellular and Molecular Biology, College of Life Science, National Taiwan University, Taipei, Taiwan.

Dear Dr. Chen,

Thank you for the submission of your manuscript to EMBO reports. We have now received the full set of referee reports that is pasted below.

As you will see, the referees acknowledge that the findings are potentially interesting. However, they also have several suggestions for how the study should be strengthened. I think all suggestions are good and should be addressed. Please let me know in case you disagree and we can discuss the exact revision requirements further, also in a video chat, if you like.

I would thus like to invite you to revise your manuscript with the understanding that the referee concerns must be fully addressed and their suggestions taken on board. Please address all referee concerns in a complete point-by-point response. Acceptance of the manuscript will depend on a positive outcome of a second round of review. It is EMBO reports policy to allow a single round of major revision only and acceptance or rejection of the manuscript will therefore depend on the completeness of your responses included in the next, final version of the manuscript.

We realize that it is difficult to revise to a specific deadline. In the interest of protecting the conceptual advance provided by the work, we recommend a revision within 3 months (24th Jul 2025). Please discuss the revision progress ahead of this time with the editor if you require more time to complete the revisions.

- 1) A data availability section providing access to data deposited in public databases is missing. If you have not deposited any data, please add a sentence to the data availability section that explains that.
- 2) Your manuscript contains statistics and error bars based on $n=2$. Please use scatter blots in these cases. No statistics should be calculated if $n=2$.

5) a complete author checklist, which you can download from our author guidelines <https://www.embopress.org/page/journal/14693178/authorguide>. Please insert information in the checklist that is also reflected in the manuscript. The completed author checklist will also be part of the RPF.

6) Please note that all corresponding authors are required to supply an ORCID ID for their name upon submission of a revised manuscript (<https://orcid.org/>). Please find instructions on how to link your ORCID ID to your account in our manuscript tracking system in our Author guidelines <https://www.embopress.org/page/journal/14693178/authorguide#authorshipguidelines>

7) Before submitting your revision, primary datasets produced in this study need to be deposited in an appropriate public

database (see <https://www.embopress.org/page/journal/14693178/authorguide#datadeposition>). Please remember to provide a reviewer password if the datasets are not yet public. The accession numbers and database should be listed in a formal "Data Availability" section placed after Materials & Method (see also <https://www.embopress.org/page/journal/14693178/authorguide#datadeposition>). Please note that the Data Availability Section is restricted to new primary data that are part of this study. * Note - All links should resolve to a page where the data can be accessed. *

12) All Materials and Methods need to be described in the main text using our 'Structured Methods' format, which is required for all research articles. According to this format, the Methods section includes a Reagents and Tools Table (listing key reagents, experimental models, software and relevant equipment and including their sources and relevant identifiers) followed by a Methods and Protocols section describing the methods using a step-by-step protocol format. The aim is to facilitate adoption of the methodologies across labs. More information on how to adhere to this format as well as a downloadable template (.docx) for the Reagents and Tools Table can be found in our author guidelines:
<https://www.embopress.org/page/journal/14693178/authorguide#structuredmethods>.

An example of a Method paper with Structured Methods can be found here: <https://www.embopress.org/doi/full/10.1038/s44320-024-00037-6#sec-4>

You are able to opt out of this by letting the editorial office know (emboreports@embo.org). If you do opt out, the Review

Process File link will point to the following statement: "No Review Process File is available with this article, as the authors have chosen not to make the review process public in this case."

I look forward to seeing a revised form of your manuscript when it is ready.

Yours sincerely,

Referee #1:

Yang et al. demonstrate that sexually dimorphic branching is regulated by both cell-autonomous mechanisms, such as cellular sex identity with integrated feeding status, and cell-non-autonomous inputs from surrounding tissues. They further show that the FOXO transcription factor DAF-16 promotes branching and its dynamic morphology by integrating nutritional cues. Lastly, the study provides evidence that both internal insulin signaling and external cues from vulval epithelial cells are required for F-actin enrichment, which facilitates the formation of the current branch structure. PVP branch morphology is highly dynamic and closely tied to physiological conditions, demonstrating a reversible and nutritionally regulated plasticity in neuronal structure.

This is a very interesting manuscript in several regards and, in principle a great candidate for publication in EMBO Reports. However, there is still some work to be done to make this a more complete story.

Major points:

1) Main experimental issue in terms of overall rigor of the study: Fig.7A (effect of sex-changed on egg-laying behavior) has several problems: First, it needs multiple independent transgenic lines and a proper control control (i.e. express something else, not fem-3. The latter is a real problem, because I understand that the same promoter that drives fem-3 is the promoter that also causes egg laying defects upon "overexpression" (presumably based on some promoter squelching effect). How can the authors exclude that the fem-3 experiment is not a reflection of a similar squelching effect.

2) The function of PVP in egg-laying behavior is not well integrated with the observation of feeding-state dependence of the branching. These two aspects need to be better explored in order to get a more satisfying view of the function of the PVP branch. Some thoughts on this:

(a) does the feeding-stage paradigm some aspect of egg-laying behavior. I could see a causal chain from feeding state controlling PVP branching to controlling egg-laying rate, but this has not been explored/considered.

(b) the authors ignore a lot of previous knowledge on egg-laying behavior. First, there's the mechanosensory aspect, mediated perhaps by uv1. Could PVP similarly be a "pressure sensor" that provides homeostatic feedback. The authors should read the uv1 literature and see how uv1-mediated egg laying behavior is controlled. And then test this with the PVPprom::fem-3 lines (once those are properly controlled; see above).

(c) similarly, the authors fail to integrate their observation with what Li and Chalfie had described in 1990 for the VC neurons. Seems like a very similar process - their branching is also controlled by epithelial cues. It's not just a question of citing this past work, the authors should rather examine the relationship of the PVP branches with the VC branches (using an nice available *ida-1::rfp* transgene). Are they in same position/associated? Do the VC branches show similar dynamics (food dependence) as PVP? The authors may want to test whether aspects of egg-laying behavior that require VCs are similar affected by PVP branches (using PVPprom::fem-3). Also, the PVP branches should be correlated with the HSN branches. These experiments will give valuable insights about the integration of the PVP branches with the rest of the well described egg-laying circuit (something completely missing from the present manuscript) and suggest further function.

Editorial changes:

kap-1 is the only marker used to assess cilia-nature of the PVP branches. Please note and cite that Christie and Koelle tested other cilia markers and apparently found no additional markers to be located there. Please discuss.

The authors must provide a more transparent description of their reagents used. For example, in Fig.1D, the authors described the localization of specific markers, such as *cla-1*, in PVP, but neither the text, the figure nor the legend describe the nature of the transgene (?) used for this study. This lack of description of reagents has to be carefully fixed throughout the entire manuscript (e.g. Fig.5D - how exactly where vulE/F killed -> state strain name in Fig legend and promoter in text). I understand and much appreciate that a strain list is provided in the MS, but the author should be more transparent.

It would also be useful to discuss the criticism that since we know from the Christie and Koelle study that transgene expression in PVP does impact branch structure, could it be that that transgenes that express specific markers (such as *cla-1*) may impact

branch structure? One could in theory solve this problem by having all these markers be expressed from a single copy, integrated array. I do not want to ask for too much, though. But I do want to see the issue discussed.

Another problem with Fig.1D is that it lacks any form of quantification. How many punctae are observed in the branch? In how many animals? The authors presumably have this data and need to show it.

Line 89: Please cite the basis for the statement of morphology and neurotransmitter identity of PVP -> White et al. 1986 and Pereira et al. 2015, eLife

Line 101: "Interestingly, we found ...". This was actually already found by Christie and Koelle. Please change to "We confirmed that..... (Christie and Koelle reference).

Line 133: A reference to the Christie and Koelle paper is also in order when the authors describe the cilia feature of these extensions.

Line 138: Ref should be Taylor et al. 2021, not Hammarlund 2018

Line 214: delete "the" before "hermaphrodite" and change "sustained" to "triggered"

In Figure 4B, the label for the X axis is confusing and hard to understand, and the y-axis label should indicate GFP::TRA-1 intensity rather than the fluorescent intensity. The representative images lack a scale bar. In the figure legends, it should be GFP::TRA-1 instead of TRA-1::GFP. Similar comments for Figure 6.

Referee #2:

Yang and colleagues identify a previously unrecognized branching of the neuron PVP in hermaphrodites, which does not occur in males. This sexually dimorphic neuronal branching event joins a growing list of unisexual structural specializations that have been identified in the *C. elegans* nervous system, such as the male specific elaborations of the sex-shared neurons PHC, DVB, PDB and PVD, to name a few (Serrano-Saiz et al 2017, Hart and Hobert 2018, Kim et al 2025, Losilevskii et al 2025, respectively), and so in and of itself is not extremely unique. The novelty comes in exposing the crosstalk between insulin signaling, neuronal branching/remodeling, and sexual context, but this could have been better tied and made more meaningful with a few additional experiments, elaborated below.

Major point:

1. How the sex-determination and the insulin pathway intersect to control branch formation has been touched on briefly but not sufficiently. The result that *daf-16* mutants suppress the PVP-masculinization phenotype is nice and points to a potential cross talk, but the even nicer experiment would be to show that reducing *tra-1* levels by masculinization in hermaphrodites affects *daf-16* activity (maybe also its levels). This can be easily done by using the *daf-16* GFP strain available from CGC and crossing it to the PVP masc transgen. This would tie up the relationship between the sex determination pathway and insulin signaling much better, and is actually a much more reasonable explanation than the puzzling experiment shown in Figure 6A, where the authors tested but didn't find (obviously, in my opinion) that the insulin pathway or *lin-39* affect *tra-1* levels.

2. The effect of insulin signaling (Figure 3) and *daf-16* seems to be different in development (branch formation) and adulthood (rod-to-wing change in response to starvation). When discussing how sexual identity impinges on insulin signaling, Fig. 4C addresses only the developmental role (i.e. *daf-16* is necessary in masculinized PVP to suppress PVP branching), but how does it affect the nutritional response in adulthood?

i. Think of an experiment to test whether the wing-to-rod nutritional response is also sexualized. If the 30% PVPmasc animals that show branching are included in the *tra-1*-low analysis (i.e. PVP is truly masculinized and yet it branches like in regular herms), the authors should starve them and check if they show wing-to-rod transformation, to understand if this phenomenon is also sexualized autonomously.

ii. What happens in males with feminized PVP when *daf-16* is gone - does PVP then grow branches (which would suggest that non-autonomous masculinized insulin signaling instructs PVP branching)?

Minor points:

1. Switch the numbering of Fig S1 and S2. S2 should come before S1, as it does in the text.

2. It should be clearly stated which sex is shown in ALL panels of all figures (FIGURE 5E for example- graph should state that *let-60* is in herms and *lin-12* is in males).

Fig. 2B and 6 - no quantification and statistics for lifeact signal? The heatmaps by themselves are not very convincing in Figure 6.

3. There is no mention in the methods of how the feminization and masculinization of PVP were done. Even more important, the *ocr-3* promoter is clearly expressed in additional cells, even if its dimmer. Please state that when describing the masculinizing /feminization experiments.

4. Fig. 4 - What is mock in Fig. 4B and C? If this is the original *tra-1::GFP* CRISPR strain, it should be called WT, not mock.

5. Figure 5E - you are missing Wild-type controls for both sexes (and also sex is not stated).

6. Cited references could include more recent papers on sexual dimorphism in *C. elegans*, for example, Laura Molina-García et al 2024, Peedikayil-Kurien et al 2025 for starvation/stress-induced sexually dimorphic plasticity, and Salzberg et al 2020 for sexually dimorphic neural circuits during neurodevelopment. See my first paragraph and make sure all the papers mentioned

there are cited (i couldn't find citation for Losilevskii et al 2025 for example).

Referee #3:

The manuscript by Yang et al. studies the sexually dimorphic collateral branching in the PVP cholinergic interneurons of *C. elegans*. They show that autonomous factors such as biological sex and non-autonomous factors such as epithelial cues from the primary vulval cells and nutritional status are all important for the formation of the collateral branch in the PVP. These cell autonomous and non-autonomous pathways converge in the remodeling of F-actin in the cytoskeleton. They use a combination of transcriptional and translational reporter, together *daf-2* and *daf-16* mutants, and sex-reversal strains to demonstrate this. Furthermore, they show that these sexually dimorphic branches in PVP are important for egg laying. The question that the authors address in this work is interesting and important to the field. They utilize a new model system in the PVP cholinergic interneurons of *C. elegans* to address their question. This is a novel system, but as the authors and other have reported (Christie and Koelle 2022), has some limitations regarding the use of genetics tools. The results are clear, but some of these need to be rigorously validated. As the authors report, expression of high-copy transgenes can result in lack of PVP branches and as such, this can serve as the explanation for some of the findings. Therefore, several of the experiments will need further controls to exclude this as a possibility. For example, in figure 4C they use sex-reversal of PVP by overexpressing *fem-3* and *tra-2(ic)*. However, the authors only show 1 line and do not have the appropriate controls to show that the transgene is not affecting/causing the phenotype. This is also an issue for the *daf-16b* rescue experiments.

Major comments:

- Throughout the text (eg line 70) and in several figures (eg Figure 1B and C, 4D and E) the authors state that PVP doesn't branch in males "branching...a feature absent in males." However, the graphs do show that close to 30% of males do have branches. The type of branches or protrusions seem to be different, but this should be clarified through the texts and graphs.
- The authors claim that PVP branches have both axonal and dendritic properties, however, there's not sufficient experimental evidence to support this claim. This is an overinterpretation of the findings and should be removed.
- The data presented in Figure S4A is not consistent with the rest of the data presented in the study. In no other context, the authors report 100% of branches in PVP, while they do in this experiment. Primary data should be reviewed, and the figure adjusted accordingly.
- The authors state that *daf-16b* is the primary isoform responsible for regulating the formation of the PVP branch, but the experiments supporting this is weak at best. Only 1 transgenic line is showed in the data, which is not standard in the field and the rescue experiments with the other isoforms was not done. Furthermore, neither panneuronal or PVP expression of *daf-16b* fully restores the lack of branching observed in the *daf-2* single mutant. This is further complicated by the fact that the PVP branching is sensitive to expression of transgenes in PVP. More experiments and controls are needed to support this result.
- The finding that the *daf-16* mutant suppressed the masculinization of PVP in hermaphrodites is very interesting, but the opposite experiment (feminization of PVP in *daf-16* males) was not done and it might be important to help clarify the interplay of insulin signaling and genetic sex.
- *daf-2* mutants show a decreased branching in PVP compared to WT and *daf-16*. However, *daf-2(e1370)* animals are developmentally delayed (see Mata-Cabana et al. 2022- <https://doi.org/10.1186/s12915-022-01295-2>). An alternative explanation is that the lack of branching observed in *daf-2* mutants is a consequence of the developmental delay (slow growth). This possibility should be excluded by looking at older *daf-2* animals (24hr older compared to the WT).
- The authors show that the PVP branches are important for egg-laying by showing that hermaphrodites with masculinized PVP lay less eggs than control hermaphrodites in a. Timeframe of 2 hours. Is it possible that this is a phenotype caused by the expression of the sex-reversal transgene? Interestingly this is the finding reported in figure S7, showing a lower # of eggs laid in 2hrs in hermaphrodites with high copy number of the *ocr-3* reporter. More controls are needed to support this finding. Additionally, further experiments should be done to clarify if the defect is within the rate of eggs laid or the total # of eggs laid.
- Many graphs and figure legends throughout the paper do not have error bars or values. This needs to be addressed as is a major issue with rigor.
- The discussion is too extensive and would benefit from focusing on the major findings of the manuscript.

Minor comments:

- The plasticity of the PVP branches is quite an interesting finding. Further experiments such as determining if these branches are present after the worm is done laying most of the eggs (day 5 or 6 of adulthood) might further support this finding.
- The timing of the refeeding experiments (how long after re-feeding was the branching scored?) is not clear in the text/methods. Please clarify.
- Font size is inconsistent throughout the text.
- Whenever possible images with red/green should be altered to be colorblind friendly. I suggest that at least the heat maps be adjusted to either a single hue or to be grayscale. See <https://www.ascb.org/diversity-equity-and-inclusion/how-to-make-scientific-figures-accessible-to-readers-with-color-blindness/> for examples.
- Graph axes would benefit from more precise labeling so they can be interpreted without having to refer to the figure legends (eg Figure S6: Fluorescence Intensity (A.U.) -> TRA-1 Fluorescence Intensity (A.U.)

Line 51-53: Sentence is not clear.

Line 98: Clarify that pdf-1 is a transcriptional reporter

Line 197: expressed primarily associated -> expressed and primarily associated

Line 213: TRA-1 is referencing the gene: so it should be tra-1.

Line 214: by a transmembrane protein TRA-2 -> by the transmembrane protein TRA-2

Line 215: by an E3-ubiquitin ligase FEM-3 -> by the E3-ubiquitin ligase FEM-3

Line 223-224: suggesting that sex identity engages in the insulin... -> Sentence is not clear.

Line 239-240: which induce apoptosis abnormally -> Sentence is not clear.

Line 325-326: Sentence is not clear.

Line 512-513: Single hermaphrodite -> A single hermaphrodite

Line 513-514 : Scored egg number -> The number of eggs laid in the plate were scored after 2 hours.

Line 678 and 681: Reference is duplicated

Line 798 and 801: Reference is duplicated

We sincerely thank the reviewers for their constructive and insightful comments and suggestions. In response, we have revised the manuscript and performed additional experiments to address the concerns raised. In particular, we have strengthened experimental rigor by adding appropriate controls (*Pocr-3::mCherry*) and generating three independent PVP^{masc} lines. We also expanded functional analyses by testing egg-laying under multiple conditions, including serotonin supplementation, and integrating PVP branching into the broader egg-laying circuit through genetic interaction experiments with *uv1* and VC neurons. In addition, we clarified the interplay between sex identity and insulin signaling, showing sexually dimorphic DAF-16 activity in PVP neurons and its role in both developmental branching and adult nutritional plasticity. Lastly, we improved transparency by clearly describing all strains, promoters, and transgenes, and by revising figure legends for clarity and acknowledged limitations of *ocr-3* promoter toxicity and suggested CRISPR-based approaches for future studies. We also replaced old figures with better representative figures. Point-by-point responses to the reviewers' comments are provided below, with **reviewer comments (R) in black** and our **responses (A) in blue**. All modifications **to the manuscript are highlighted in red**. Taken together, we believe these revisions have significantly substantiated our manuscript and that it now meets the high standards of *EMBO Reports*.

Responses to reviewers

Referee #1:

(R) Yang et al. demonstrate that sexually dimorphic branching is regulated by both cell-autonomous mechanisms, such as cellular sex identity with integrated feeding status, and cell-non-autonomous inputs from surrounding tissues. They further show that the FOXO transcription factor DAF-16 promotes branching and its dynamic morphology by integrating nutritional cues. Lastly, the study provides evidence that both internal insulin signaling and external cues from vulval epithelial cells are required for F-actin enrichment, which facilitates the formation of the current branch structure. PVP branch morphology is highly dynamic and closely tied to physiological conditions, demonstrating a reversible and

nutritionally regulated plasticity in neuronal structure.

This is a very interesting manuscript in several regards and, in principle a great candidate for publication in EMBO Reports. However, there is still some work to be done to make this a more complete story.

(A) We thank the reviewer for their insightful comments and constructive suggestions. In response, we have substantially revised the manuscript and incorporated new behavioral data that further support and strengthen our main conclusions.

(R) Major points:

1) Main experimental issue in terms of overall rigor of the study: Fig.7A (effect of sex-changed on egg-laying behavior) has several problems: First, it needs multiple independent transgenic lines and a proper control control (i.e. express something else, not *fem-3*). The latter is a real problem, because I understand that the same promoter that drives *fem-3* is the promoter that also causes egg laying defects upon "overexpression" (presumably based on some promoter squelching effect). How can the authors exclude that the *fem-3* experiment is not a reflection of a similar squelching effect.

(A) We thank the reviewer for highlighting this important concern regarding the behavioral data. To address these issues, we used *Pocr-3::mCherry* (7.5 ng/μl), thereafter called PVP^{mCherry}, as a control, producing ~80–85% PVP branching, and generated three independent lines of PVP masculinization (PVP^{masc}) lines. Consistently, PVP^{mCherry} only showed a slight reduction of egg-laying rates, whereas three independent PVP^{masc} lines displayed a marked reduction of egg-laying rate, albeit the basal line for egg-laying rate was variable across experiments. We reason the variability is probably due to the perturbations of environmental factors, such as temperatures. Thus, we utilized a liquid-based assay developed by the Koelle lab with/without 18.5 mM serotonin. We found the same conclusion from the isolated system. Thus, our results demonstrate that the observed egg-laying defects are specific to PVP masculinization rather than a nonspecific effect of the *ocr-*

3 promoter. We have added these results to Figure 7A and Figure 7E and revised the results as below:

Line 335-342

...We found that hermaphrodites with masculinized PVP neurons, which lacked collateral branching, exhibited significantly reduced egg-laying rates, while expression of mCherry had minimal effect (Fig. 7A). The effect appears to be specific to egg-laying behavior, as fecundity was not substantially influenced by the sexual state of PVP neurons (Fig. EV3A). Notably, restoring PVP branching via *daf-16* mutations largely rescued the egg-laying defects caused by PVP masculinization (Fig. 7A). Together, these results indicate that PVP branch formation is essential for normal egg-laying behavior...

Line 351-359

...To probe their relationship with serotonin signaling, we exposed D1 hermaphrodites to exogenous serotonin in liquid buffer (Collins *et al*, 2016). As expected, while egg laying was attenuated in the absence of food, wild-type hermaphrodites still laid eggs when supplied with serotonin. By contrast, PVP^{masc} hermaphrodites lacking branches showed a reduced egg-laying rate even in the presence of serotonin (Fig. 7E), supporting that PVP branches act downstream or parallel of HSN neurons. Collectively, these findings suggest that PVP branches contribute to the modulation of egg-laying circuitry downstream of serotonergic pathways critical for reproduction in hermaphrodites...

(R) 2) The function of PVP in egg-laying behavior is not well integrated with the observation of feeding-state dependence of the branching. These two aspects need to be better explored in order to get a more satisfying view of the function of the PVP branch. Some thoughts on this:

(a) does the feeding-stage paradigm some aspect of egg-laying behavior. I could see a causal chain from feeding state controlling PVP branching to controlling egg-laying rate, but this has not been explored/considered.

(A) We thank the reviewer for this constructive suggestion. To strengthen the link between nutritional status, PVP branching, and egg-laying behavior, we asked whether the egg-laying defects in the PVP^{masc} hermaphrodites can be rescued by the *daf-16* mutations. We found that restoration of PVP branching by the *daf-16* mutations in the PVP^{masc} hermaphrodites completely rescued the egg-laying defects, reinforcing the link between sex identity, insulin signaling, and egg-laying

behavior. The new data was incorporated into Figure 7A and main text was modified as shown in the point 1 (line 335-342).

Interestingly, we found that *daf-16* mutations cannot suppress the branch-loss and egg-laying defects caused by high-copy transgene, suggesting that transgenic toxicity disrupts branch development by a distinct pathway. We thus removed this data from our main text and original Figure S7 to adhere to our mechanistic analysis and avoid any confusions from readers.

(R) (b) the authors ignore a lot of previous knowledge on egg-laying behavior. First, there's the mechanosensory aspect, mediated perhaps by *uv1*. Could PVP similarly be a "pressure sensor" that provides homeostatic feedback. The authors should read the *uv1* literature and see how *uv1*-mediated egg laying behavior is controlled. And then test this with the *PVPprom::fem-3* lines (once those are properly controlled; see above).

(A) We thank the reviewer for this valuable suggestion. We agree that PVP branches could, in principle, serve a mechanosensory role analogous to *uv1* cells, although their effects on egg-laying appear to be opposite. *uv1* neuroendocrine cells act as mechanosensors that release tyramine to inhibit HSN activity via the *lgc-55* tyramine-gated chloride channel; disruption of this *uv1*-HSN axis leads to premature egg laying and increased egg-laying rates.

We have incorporated this background into the results and discussion to outline a plausible mechanosensory role for PVP branches. To experimentally test the interaction between the uv1-HSN axis and PVP neurons, we examined genetic interactions between *lgc-55* mutants and PVP^{masc} hermaphrodites. Masculinization of PVP neurons in the *lgc-55* mutant background reduced egg laying rate. These findings suggest that PVP neurons are unlikely to act in the same uv1-HSN axis to control egg-laying behavior.

We have integrated the new data into Figure 7D and revised our manuscript accordingly as below:

Line 343-351

...In *C. elegans*, serotonin released from HSN neurons initiates the active phase of egg laying by sensitizing vulval muscles to cholinergic input from VC neurons (Fig. 7B) (Garcia & Portman, 2016). Neuroendocrine uv1 cells modulate the egg-laying circuit by releasing tyramine, which suppresses HSN activity through the tyramine-gated chloride channel LGC-55, thereby terminating the active phase (Fig. 7B) (Yan *et al*, 2025). Strikingly, we found no physical contacts between PVP branches and HSN or VC neurons (Fig. 7C), suggesting that PVP branches modulate egg laying indirectly. Loss of PVP branches in PVP^{masc} hermaphrodites further reduced egg-laying rates in the *lgc-55* mutant (Fig. 7D), indicating that PVP branches act in parallel to the uv1-HSN axis...

(R) (c) similarly, the authors fail to integrate their observation with what Li and Chalfie had described in 1990 for the VC neurons. Seems like a very similar process - their branching is also controlled by epithelial cues. It's not just a question of citing this past work, the authors should rather examine the relationship of the PVP branches with the VC branches (using an nice available *ida-1::rfp* transgene). Are they in same position/associated? Do the VC branches show similar dynamics (food dependence) as PVP? The authors may want to test whether aspects of egg-laying behavior that require VCs are similar affected by PVP branches (using PVP_{prom}::*fem3*-). Also, the PVP branches should be correlated with the HSN branches. These experiments will give valuable insights about the integration of the PVP branches with the rest of the well described egg-laying circuit (something completely missing from the present manuscript) and suggest further function.

(A) We thank the reviewer for the nice suggestions. Using GFP driven by *nlp-3* and *cat-1* to label HSN and VC neurons, respectively, we found that neither HSN nor VC branches physically attached to PVP branches by confocal imaging, suggesting that PVP branches are structurally distinct from both VC and HSN neurite branches. We have integrated the new data into main text and Figure 7C.

We also examined VC branch dynamics under different nutritional states and found that VC neurons maintain rod-like structures regardless of feeding status (comparing Figure 7C and below), in contrast to the nutritional plasticity of PVP branches as shown below:

Functionally, following the established liquid serotonin assay used for studying the role of VC neurons under the serotonin-mediated (18.5 mM) egg-laying behaviors, masculinization of PVP neurons significantly reduced egg laying rate, suggesting that PVP branches act downstream or parallel of the serotonergic pathway similar to VC neurons in the prior paper (Collins *et al.*, 2016). This result was integrated into the Figure 7E.

In addition to new data, we also modified our results and discussions to integrate the role of PVP branches into the broader egg-laying circuit framework, with key references cited (Li&Chalfie, 1990). Overall, we believe that these results provide

valuable information regarding the functional role of PVP branches in the egg-laying circuit.

In results (line 343-359):

In *C. elegans*, serotonin released from HSN neurons initiates the active phase of egg laying by sensitizing vulval muscles to cholinergic input from VC neurons (Fig. 7B) (Garcia & Portman, 2016). Neuroendocrine uv1 cells modulate the egg-laying circuit by releasing tyramine, which suppresses HSN activity through the tyramine-gated chloride channel LGC-55, thereby terminating the active phase (Fig. 7B) (Yan *et al.*, 2025). Strikingly, we found no physical contacts between PVP branches and HSN or VC neurons (Fig. 7C), suggesting that PVP branches modulate egg laying indirectly. Loss of PVP branches in PVP^{masc} hermaphrodites further reduced egg-laying rates in the *lgc-55* mutant (Fig. 7D), indicating that PVP branches act in parallel to the uv1-HSN axis. To probe their relationship with serotonin signaling, we exposed D1 hermaphrodites to exogenous serotonin in liquid buffer (Collins *et al.*, 2016). As expected, while egg laying was attenuated in the absence of food, wild-type hermaphrodites still laid eggs when supplied with serotonin. By contrast, PVP^{masc} hermaphrodites lacking branches showed a reduced egg-laying rate even in the presence of serotonin (Fig. 7E), supporting that PVP branches act downstream or parallel of HSN neurons. Collectively, these findings suggest that PVP branches contribute to the modulation of egg-laying circuitry downstream of serotonergic pathways critical for reproduction in hermaphrodites.

In discussion (line 388-399):

...Egg-laying behaviors are mainly controlled by HSN motor neurons and other surrounding VC neurons and uv1 neuroendocrine cells (Collins *et al.*, 2016; Yan *et al.*, 2025). Our data indicate that PVP branches modulate egg-laying behaviors presumably via peptidergic signaling. Interestingly, we did not observe physical contact between PVP branches, HSN, and VC neurons, supporting the view that extrasynaptic signaling is involved. While our results demonstrated the role of PVP branches in egg-laying circuits, the exact function of PVP branches remains unknown. Recently, mechanical sensation in VC, vulva muscles, and uv1 neuroendocrine cells serve as a feedback loop to coordinate egg-laying behaviors (Li & Chalfie, 1990; Medrano & Collins, 2023; Yan *et al.*, 2025). One tentative idea is that PVP branches also sense the mechanical force in this module during egg-laying to coordinate locomotion. However, more analyses are needed to substantiate this hypothesis in the future...

(R) Editorial changes:

kap-1 is the only marker used to assess cilia-nature of the PVP branches. Please

note and cite that Christie and Koelle tested other cilia markers and apparently found no additional markers to be located there. Please discuss.

(A) We have emphasized the findings from the preprint of Christie and Koelle in results and acknowledged the limitation in discussion. Because not all PVP branches accommodated the KAP-1 (82.5% of PVP branches are KAP-1 positive), we decided to tone down our initial expression as below:

In abstract (line 6-7):

PVP branches form near the vulva and exhibit dynamic morphologies enriched with synaptic proteins for dense core vesicles but not synaptic vesicles, suggesting a role in selective neuropeptide transmission.

In results (line 142-147):

We found that *kap-1*, which encodes a component of the kinesin complex involved in intraflagellar transport, is expressed in PVP neurons. KAP-1 labeling revealed strong enrichment in the cell soma and the branch tip (Fig. 1E and Appendix Fig. 1C). Notably, Christie and Koelle (2022) did not detect other ciliary markers in PVP branches, leaving the biological function of KAP-1 in these structures unresolved (Christie & Koelle, 2022).

We noted that Christie and Koelle only briefly mentioned that they tested a few transgenes but did not find any colocalization with PVP neurons. However, the exact content of specific markers was not provided in BioRxiv (File S1). Thus, we could not compare the differences between our findings and theirs.

(R) The authors must provide a more transparent description of their reagents used. For example, in Fig.1D, the authors described the localization of specific markers, such as *cla-1*, in PVP, but neither the text, the figure nor the legend describe the nature of the transgene (?) used for this study. This lack of description of reagents has to be carefully fixed throughout the entire manuscript (e.g. Fig.5D - how exactly where vulE/F killed -> state strain name in Fig legend and promoter in text). I understand and much appreciate that a strain list is provided in the MS, but the author should be more transparent.

(A) We apologize for the ambiguous description in term of methodology. We have modified our descriptions to clearly notify readers the background and the description of reagents and methods. First, all materials are listed in the reagent and tools table. Second, we also modified all figure legends to label transgenes used in each figure. We believe that these changes significantly fostered the transparency of the use of materials for readers.

(R) It would also be useful to discuss the criticism that since we know from the Christie and Koelle study that transgene expression in PVP does impact branch structure, could it be that that transgenes that express specific markers (such as cla-1) may impact branch structure? One could in theory solve this problem by having all these markers be expressed from a single copy, integrated array. I do not want to ask for too much, though. But I do want to see the issue discussed.

(A) We thank reviewer for this great comment. We have added a paragraph in discussion to reveal the potential limitations in this study.

In discussion (line 443-453):

Limitations of This Study

Our phenotypic analyses revealed a transgenic toxicity associated with the *ocr-3* promoter (PVP-specific) that affects PVP branch morphogenesis, consistent with previous observations (Christie & Koelle, 2022). Although we minimized this effect by using a relatively low transgene concentration (7.5 ng/μL), a mild reduction in PVP branch formation and wing-to-rod transitions was still observed. Consequently, any manipulation of gene activity or expression of synaptic or cytoskeletal markers under the *ocr-3* promoter is likely to partially alter the normal morphology and function of PVP neurons. To address this limitation, future studies could employ CRISPR-mediated insertion of a single-copy transgene driven by the *ocr-3* promoter, or identify alternative PVP-specific promoters with minimal toxicity for labeling and functional assays.

(R) Another problem with Fig.1D is that it lacks any form of quantification. How many punctae are observed in the branch? In how many animals? The authors presumably have this data and need to show it.

(A) We have quantified the percentage of animals carrying each marker and labeled it along with our images in Figure 1. The quantification of the number of puncta was provided in the figure legend.

(R) Line 89: Please cite the basis for the statement of morphology and neurotransmitter identity of PVP -> White et al. 1986 and Pereira et al. 2015, eLife

Line 101: "Interestingly, we found ...". This was actually already found by Christie and Koelle. Please change to "We confirmed that..... (Christie and Koelle reference).

Line 138: Ref should be Taylor et al. 2021, not Hammarlund 2018

Line 214: delete "the" before "hermaphrodite" and change "sustained" to "triggered"

In Figure 4B, the label for the X axis is confusing and hard to understand, and the y-axis label should indicate GFP::TRA-1 intensity rather than the fluorescent intensity. The representative images lack a scale bar. In the figure legends, it should be GFP::TRA-1 instead of TRA-1::GFP. Similar comments for Figure 6.

(A) We have corrected it in our revised manuscript.

(R) Line 133: A reference to the Christie and Koelle paper is also in order when the authors describe the cilia feature of these extensions.

(A) We have cited the paper as shown below (line 137-139):

While our initial characterization suggested axonal features, previous study also suggests that wing- and rod-like structures of PVP branches resemble dendritic structures in sensory neurons (Christie & Koelle, 2022).

Referee #2:

(R) Yang and colleagues identify a previously unrecognized branching of the

neuron PVP in hermaphrodites, which does not occur in males. This sexually dimorphic neuronal branching event joins a growing list of unisexual structural specializations that have been identified in the *C. elegans* nervous system, such as the male specific elaborations of the sex-shared neurons PHC, DVB, PDB and PVD, to name a few (Serrano-Saiz et al 2017, Hart and Hobert 2018, Kim et al 2025, Losilevskii et al 2025, respectively), and so in and of itself is not extremely unique. The novelty comes in exposing the crosstalk between insulin signaling, neuronal branching/remodeling, and sexual context, but this could have been better tied and made more meaningful with a few additional experiments, elaborated below.

(A) We thank the reviewer for many insightful suggestions and comments. We have revised our manuscript and added behavioral data to substantiate the major conclusion in this paper.

(R) Major point:

1. How the sex-determination and the insulin pathway intersect to control branch formation has been touched on briefly but not sufficiently. The result that *daf-16* mutants suppress the PVP-masculinization phenotype is nice and points to a potential cross talk, but the even nicer experiment would be to show that reducing *tra-1* levels by masculinization in hermaphrodites affects *daf-16* activity (maybe also its levels). This can be easily done by using the *daf-16* GFP strain available from CGC and crossing it to the PVP masc transgen. This would tie up the relationship between the sex determination pathway and insulin signaling much better, and is actually a much more reasonable explanation than the puzzling experiment shown in Figure 6A, where the authors tested but didn't find (obviously, in my opinion) that the insulin pathway or *lin-39* affect *tra-1* levels.

(A) We thank reviewer for the great suggestion. Our results suggest that DAF-16 activity is sexually dimorphic in PVP neurons, with higher DAF-16 activity in males than in hermaphrodites. To assess this, we first attempted to visualize endogenous DAF-16 using *ot853* (*daf-16::mNeon*) and the translational reporter *zIs356* (*Pdaf-16::daf-16a/b*), but detected no fluorescence in PVP neurons, consistent with low expression levels in CeNGEN data. To circumvent this, we expressed *daf-16b::mCherry* under the *ocr-3* promoter and observed significantly

stronger DAF-16::mCherry signal in males than in hermaphrodites. We attempted to confirm this by co-expressing *Pocr-3::FEM-3::SL2::BFP*, but were unable to recover animals with consistent co-expression probably due to the transgenic toxicity. Although we could not achieve stable co-expression despite multiple attempts, our observations of sexually dimorphic DAF-16 activity still support the conclusion. We have integrated the new data into Figure 3D and modified results and discussions accordingly:

In results (line 219-224):

...Collectively, our data support a model in which elevated *daf-16b* activity in PVP neurons drives branch loss in males. To test whether this reflects sex-specific *daf-16* activity, we expressed DAF-16b::mCherry in PVP neurons using the *ocr-3* promoter and compared fluorescence between males and hermaphrodites. Males exhibited higher DAF-16b::mCherry levels, supporting a role for sexually dimorphic DAF-2/DAF-16 signaling in PVP branch formation....

In discussion (line 421-429):

The Interplay Between Biological Sex and the Insulin Pathway

Our data suggest that biological sex regulates PVP branching through the *daf-16*/FOXO transcription factor, positioning the insulin pathway downstream of biological sex. In parallel to our DAF-16 imaging experiments, we examined whether the insulin pathway or *lin-39* could modulate *tra-1* levels in PVP neurons (Fig. 6A). While no significant changes were detected, these results serve as complementary observations rather than the primary evidence for sex-insulin pathway interaction. Our imaging data directly support a model in which DAF-16 activity is sexually dimorphic, with higher levels in males, and likely contributes to the branching differences between sexes.

(R) 2. The effect of insulin signaling (Figure 3) and *daf-16* seems to be different in development (branch formation) and adulthood (rod-to-wing change in response to starvation). When discussing how sexual identity impinges on insulin signaling, Fig. 4C addresses only the developmental role (i.e. *daf-16* is necessary in masculinized PVP to suppress PVP branching), but how does it affect the nutritional response in adulthood?

(A) We thank reviewer for this suggestion. To address the role of *daf-16* for wing-rod transition, we studied whether the dynamic changes are resumed in the *daf-16*; *Pocr-3::daf-16b* under the starvation experimental paradigm. Similar to the role in

development, *daf-16* acts autonomously in PVP neurons to control wing-rod transition. We have integrated this data into Figure 3A and expand our original main text as below:

Line 215-218

...Beyond branch initiation, expression of *daf-16b* in PVP neurons also rescued the wing-to-rod transition in the *daf-16* mutant (Fig. 3A), suggesting a cell-autonomous role for *daf-16b* in regulating morphological dynamics in response to nutritional cues...

(R) i. Think of an experiment to test whether the wing-to-rod nutritional response is also sexualized. If the 30% PVP^{masc} animals that show branching are included in the tra-1-low analysis (i.e. PVP is truly masculinized and yet it branches like in regular herms), the authors should starve them and check if they show wing-to-rod transformation, to understand if this phenomenon is also sexualized autonomously.

(A) We thank reviewer for this suggestion. In PVP^{masc} hermaphrodites that retained PVP branches, only $\leq 10\%$ exhibited wing-like structures under well-fed conditions, compared to $>45\%$ in wild-type hermaphrodites. This suggests that a masculinized sex identity markedly reduces the formation of wing-like branches even in the absence of starvation, highlighting the autonomous role of sexual identity in shaping PVP morphology. We have added this observation to the main text as below:

Line 240-244

...indicating that the hermaphroditic fate of PVP neurons is required for branch formation. Strikingly, branch loss in PVP^{masc} hermaphrodites was restored by the *daf-16* mutation, indicating that *daf-16* functions downstream of sexual identity to control PVP branching (Fig. 4C). In addition, PVP^{masc} hermaphrodites rarely formed wing-like branches even in well-fed condition, suggesting that masculinization limits the nutritional plasticity of PVP morphology (Fig. EV2D)...

(R) ii. What happens in males with feminized PVP when *daf-16* is gone - does PVP then grow branches (which would suggest that non-autonomous masculinized insulin signaling instructs PVP branching)?

(A) We thank reviewer for pointing this out. To test it, we generated *daf-16; Pocr-3::TRA-2(IC)* and examined PVP branching in males. We found that feminization of PVP neurons in the *daf-16* males did not promote PVP branching. This data is consistent with the autonomous role of *daf-16*. We have integrated this data into Figure 4D and modified our results:

Line 244-248

...Conversely, feminization of PVP neurons (PVP^{fem}) in males restored TRA-1 activity but did not promote branch formation even with the *daf-16* mutations (Fig. 4B and D). These results demonstrate that while the hermaphroditic identity of PVP neurons is necessary for....

(R) Minor points:

1. Switch the numbering of Fig S1 and S2. S2 should come before S1, as it does in the text.

(A) We apologize for causing the confusion. We have moved original Fig S2A to the new Fig S1 to better fit the flow and then corrected all the labeling and text in figure legends.

2. It should be clearly stated which sex is shown in ALL panels of all figures (FIGURE 5E for example- graph should state that *let-60* is in herms and *lin-12* is in males).

(A) We have modified all figures and clearly labeled sex identity.

Fig. 2B and 6 - no quantification and statistics for lifeact signal? The heatmaps by themselves are not very convincing in Figure 6.

(A) We have quantified LifeAct signal by comparing the average of signal intensity in vulva region to non-vulva region. This data shows an enrichment of LifeAct signal near vulva in wild-type but this enrichment was absent in early developmental stages or in *daf-2/lin-39* mutants. We have incorporated this data into Figure 4 and Figure 6 next to the heatmaps.

3. There is no mention in the methods of how the feminization and masculinization

of PVP were done. Even more important, the *ocr-3* promoter is clearly expressed in additional cells, even if its dimmer. Please state that when describing the masculinizing /feminization experiments.

(A) We have modified our method section to improve the clarity. Changes are shown below:

line 507-512

Masculinization and feminization of PVP neurons

PVP neuron masculinization was induced by overexpressing *fem-3*, which encodes an E3 ubiquitin ligase that targets the transcription factor TRA-1 for degradation, using the transgene *chcEx027(Pocr-3::FEM-3::SL2::mCherry)*. Feminization was achieved by overexpressing the intracellular cytosolic fragment (IC) of *tra-2* using the transgene *chcEx009(Pocr-3::TRA-2::SL2::mCherry)*.

Regarding the *ocr-3* promoter, our transgenic analysis indicates that it is only expressed in PVP neurons as shown in the Figure 1A. To avoid the confusion from the use of other promoters (such as the *pdf-1* promoter), we have labeled the use of each promoter in results to clarify this issue.

(R) 4. Fig. 4 - What is mock in Fig. 4B and C? If this is the original *tra-1::GFP* CRISPR strain, it should be called WT, not mock.

(A) We thank reviewer for raising this question. The mock represents the transgenic animals carrying *Pocr-3::mCherry* as a control. We have added the labeling to avoid confusions.

5. Figure 5E - you are missing Wild-type controls for both sexes (and also sex is not stated).

(A) We have included wild-type controls for both sexes in Figure 5E and labeled the sex identity.

(R) 6. Cited references could include more recent papers on sexual dimorphism in *C. elegans*, for example, Laura Molina-García et al 2024, Peedikayil-Kurien et al 2025 for starvation/stress-induced sexually dimorphic plasticity, and Salzberg et al 2020 for sexually dimorphic neural circuits during neurodevelopment. See my first

paragraph and make sure all the papers mentioned there are cited (i couldn't find citation for Losilevskii et al 2025 for example).

(A) We have cited the paper to give a more complete view of sexually dimorphic neurodevelopment in *C. elegans* in introduction. Because the Barrios paper primarily focuses on sexual conditioning for valence, we did not cite this paper in our revised manuscript.

Referee #3:

The manuscript by Yang et al. studies the sexually dimorphic collateral branching in the PVP cholinergic interneurons of *C. elegans*. They show that autonomous factors such as biological sex and non-autonomous factors such as epithelial cues from the primary vulval cells and nutritional status are all important for the formation of the collateral branch in the PVP. These cell autonomous and non-autonomous pathways converge in the remodeling of F-actin in the cytoskeleton. They use a combination of transcriptional and translational reporter, together *daf-2* and *daf-16* mutants, and sex-reversal strains to demonstrate this. Furthermore, they show that these sexually dimorphic branches in PVP are important for egg laying. The question that the authors address in this work is interesting and important to the field. They utilize a new model system in the PVP cholinergic interneurons of *C. elegans* to address their question. This is a novel system, but as the authors and other have reported (Christie and Koelle 2022), has some limitations regarding the use of genetics tools. The results are clear, but some of these need to be rigorously validated. As the authors report, expression of high-copy transgenes can result in lack of PVP branches and as such, this can serve as the explanation for some of the findings. Therefore, several of the experiments will need further controls to exclude this as a possibility. For example, in figure 4C they use sex-reversal of PVP by overexpressing *fem-3* and *tra-2(ic)*. However, the authors only show 1 line and do not have the appropriate controls to show that the transgene is not affecting/causing the phenotype. This is also an issue for the *daf-16b* rescue experiments.

(A) We thank the reviewer for many insightful suggestions and comments. We have revised our manuscript and added several controls to substantiate the major conclusion in this paper.

Major comments:

- Throughout the text (eg line 70) and in several figures (eg Figure 1B and C, 4D and E) the authors state that PVP doesn't branch in males "branching...a feature absent in males." However, the graphs do show that close to 30% of males do have branches. The type of branches or protrusions seem to be different, but this should be clarified through the texts and graphs.

(A) We thank the reviewer for this nice suggestion. We indeed think that protrusions in males are morphologically and functionally distinct to PVP branches in hermaphrodites. To better distinguish these two, we rescored all male data by placing stringer criteria (process length $> 5 \mu\text{m}$ to $>8 \mu\text{m}$). The new criteria allow us to better separate the protrusion and mature branch in males and hermaphrodites. We have modified Figure 1B, 1C, Figure EV1D, and Figure 4D with the new criteria.

In methods (line 522-529):

Quantification of PVP branching:

30-50 D1 adult animals were quantified by picked L4 larva a day before scoring. Animals were then mounted on a 5% agarose pad and paralyzed with 1% sodium azide. Images were taken by ZEN Blue 3.6 software using 63X objective in a Zeiss Imager M2 microscope with an AxioCam 705 Mono camera. We defined a mature branch as one that has a protrusion of longer than 8 μm . The percentage of PVP branching in a population was subsequently scored.

(R) - The authors claim that PVP branches have both axonal and dendritic properties, however, there's not sufficient experimental evidence to support this claim. This is an overinterpretation of the findings and should be removed.

(A) We agree with the reviewer that our initial statement might be overstated. We have removed the description and toned down the role of PVP branches as axonal or dendrites as shown below:

In Abstract (line 6-7):

PVP branches form near the vulva and exhibit dynamic morphologies enriched with synaptic proteins for dense core vesicles but not synaptic vesicles, suggesting a role in selective neuropeptide transmission.

In Introduction (line 76-77):

In addition, mature branches also accommodated cilia proteins, although their functional role remains uncertain.

In Results (line 145-147):

Notably, Christie and Koelle (2022) did not detect other ciliary markers in PVP branches, leaving the biological function of KAP-1 in these structures unresolved (Christie & Koelle, 2022).

(R)- The data presented in Figure S4A is not consistent with the rest of the data presented in the study. In no other context, the authors report 100% of branches in PVP, while they do in this experiment. Primary data should be reviewed, and the figure adjusted accordingly.

(A) We appreciate that the reviewer points this out and apologize for causing the confusion. In our paper, we used Mock to represent *chcSil* carrying *Pocr-3::mCherry* as a control whenever we are comparing the effect of masculinization or feminization. In this case, PVP branching rate was around 85-90% due to the transgenic toxicity. However, PVP branching rate in the *chcSil* without transgene was 98-100% as shown in the Figure 1C. We also have reviewed our raw data and redo the scoring again, and we observed over 98% of PVP branching in different nutritional conditions. Therefore, we would like to retain the original data in Figure S4A (now Figure EV2).

(R) - The authors state that *daf-16b* is the primary isoform responsible for regulating the formation of the PVP branch, but the experiments supporting this is weak at best. Only 1 transgenic line is showed in the data, which is not standard in the field and the rescue experiments with the other isoforms was not done. Furthermore, neither panneuronal or PVP expression of *daf-16b* fully restores the lack of branching observed in the *daf-2* single mutant. This is further complicated by the fact that the PVP branching is sensitive to expression of transgenes in PVP. More experiments and controls are needed to support this result.

(A) We thank the reviewer for this constructive suggestion. To address the concern, we generated two additional independent *Pocr-3::daf-16b* transgenic lines in the *daf-16; daf-2* mutant background. All three lines showed consistent restorations of the branching defect, with one line restoring branching almost to the *daf-2* mutant levels. In contrast, three independent lines of *Pocr-3::daf-16f* did not restore branching, indicating that this phenotype is specific to isoform b. Our new data thus confirm that *daf-16b* is the isoform responsible for regulating PVP branch formation. We have integrated the new data into Figure 3C and modified our results as shown below:

Line 210-218

...while driving *daf-16b* expression specifically in PVP neurons recapitulated the *daf-2* mutant phenotype (Fig. 3C) (Christensen *et al.*, 2011). This effect was specific to the *daf-16b* isoform, as expression of *daf-16f* in PVP neurons failed to suppress branching in the *daf-16; daf-2* double mutant (Fig. 3C). These results indicate that *daf-16b* acts cell-autonomously within PVP neurons to inhibit branch formation. Beyond branch initiation, expression of *daf-16b* in PVP neurons also rescued the wing-to-rod transition in the *daf-16* mutant (Fig. 3A), suggesting a cell-autonomous role for *daf-16b* in regulating morphological dynamics in response to nutritional cues...

(R) - The finding that the *daf-16* mutant suppressed the masculinization of PVP in hermaphrodites is very interesting, but the opposite experiment (feminization of PVP in *daf-16* males) was not done and it might be important to help clarify the interplay of insulin signaling and genetic sex.

(A) We thank reviewer for this suggestion. To test it, we generated *daf-16; Pocr-3::TRA-2(IC)* and examined PVP branching in males. We found that feminization of PVP neurons in the *daf-16* males did not promote PVP branching. This data is consistent with the autonomous role of *daf-16*. We have integrated this data into Figure 4D and modified our results as below:

Line 244-247

..Conversely, feminization of PVP neurons (PVP^{fem}) in males restored TRA-1 activity but did not promote branch formation even with the *daf-16* mutations (Fig. 4B and D). These results demonstrate that while the hermaphroditic identity of PVP neurons is necessary for...

(R)- *daf-2* mutants show a decreased branching in PVP compared to WT and *daf-16*. However, *daf-2(e1370)* animals are developmentally delayed (see Mata-Cabana et al. 2022- <https://doi.org/10.1186/s12915-022-01295-2>). An alternative explanation is that the lack of branching observed in *daf-2* mutants is a consequence of the developmental delay (slow growth). This possibility should be excluded by looking at older *daf-2* animals (24hr older compared to the WT).

(A) We thank the reviewer for raising this concern. We have tested the *daf-2* mutants in D2 adults, and we found that the PVP branching phenotype was not altered. We have incorporated this control into Figure S4C and mentioned the data in the main text as below:

Line 192-195

Loss of *daf-2* significantly reduced PVP branching, while the *daf-16* mutation had a limited effect (Fig. 3B). We confirmed that the branch loss persisted in the D2 *daf-2* mutant, indicating that the branch loss is not due to the delayed development of mutant hermaphrodites (Fig. EV2C) (Mata-Cabana et al, 2022).

(R) - The authors show that the PVP branches are important for egg-laying by

showing that hermaphrodites with masculinized PVP lay less eggs than control hermaphrodites *ina*. Timeframe of 2 hours. Is it possible that this is a phenotype caused by the expression of the sex-reversal transgene? Interestingly this is the finding reported in figure S7, showing a lower # of eggs laid in 2hrs in hermaphrodites with high copy number of the *ocr-3* reporter. More controls are needed to support this finding. Additionally, further experiments should be done to clarify if the defect is within the rate of eggs laid or the total # of eggs laid.

(A) We thank the reviewer for highlighting this important concern regarding the behavioral data. Our results demonstrate that the observed egg-laying defects are specific to PVP masculinization rather than a nonspecific effect of the *ocr-3* promoter. As a control, we used *Pocr-3::mCherry* (7.5 ng/μl), which produced ~80–85% PVP branching but caused only a mild reduction in egg laying compared to non-transgenic animals. This rate remained significantly higher than that in PVP^{masc} hermaphrodites. We tested three independent lines and obtained the consistent effect, while the basal line for egg-laying numbers was variable. To better control variability of behavioral data, we restored PVP branches caused by PVP^{masc}. Introducing *daf-16* mutations abolished the egg-laying reduction in PVP^{masc} hermaphrodites, further supporting a causal role for PVP branching. Using a liquid serotonin assay (Koelle lab method), we obtained the same conclusion as on solid agar. Thus, these additional controls confirm that the observed phenotype is specific to PVP masculinization and not a promoter artifact, strengthening the causal link between branching and egg-laying behavior. these data also reinforce the link between sex identity, insulin signaling, and egg-laying behavior. We have added these results to Figure 7A, D, E and revised the main text as below:

Line 335-359

...We found that hermaphrodites with masculinized PVP neurons, which lacked collateral branching, exhibited significantly reduced egg-laying rates, while expression of mCherry had minimal effect (Fig. 7A). The effect appears to be specific to egg-laying behavior, as fecundity was not substantially influenced by the sexual state of PVP neurons (Fig. EV3A). Notably, restoring PVP branching via *daf-16* mutations largely rescued the egg-laying defects caused by PVP masculinization (Fig. 7A). Together, these results indicate that PVP branch formation is essential for normal egg-laying behavior.

In *C. elegans*, serotonin released from HSN neurons initiates the active phase of egg laying by sensitizing vulval muscles to cholinergic input from VC neurons (Fig. 7B) (Garcia & Portman, 2016). Neuroendocrine uv1 cells modulate the egg-laying circuit by releasing tyramine, which suppresses HSN activity through the tyramine-gated chloride channel LGC-55, thereby terminating the active phase (Fig. 7B) (Yan *et al.*, 2025). Strikingly, we found no physical contacts between PVP branches and HSN or VC neurons (Fig. 7C), suggesting that PVP branches modulate egg laying indirectly. Loss of PVP branches in PVP^{masc} hermaphrodites further reduced egg-laying rates in the *lgc-55* mutant (Fig. 7D), indicating that PVP branches act in parallel to the uv1-HSN axis. To probe their relationship with serotonin signaling, we exposed DI hermaphrodites to exogenous serotonin in liquid buffer (Collins *et al.*, 2016). As expected, while egg laying was attenuated in the absence of food, wild-type hermaphrodites still laid eggs when supplied with serotonin. By contrast, PVP^{masc} hermaphrodites lacking branches showed a reduced egg-laying rate even in the presence of serotonin (Fig. 7E), supporting that PVP branches act downstream of HSN neurons. Collectively, these findings suggest that PVP branches contribute to the modulation of egg-laying circuitry downstream of serotonergic pathways critical for reproduction in hermaphrodites.....

Interestingly, we found that *daf-16* mutations cannot suppress the branch-loss and egg-laying defects caused by high-copy transgene, suggesting that transgenic toxicity disrupts branch development in a separate manner. We thus removed this data from our main text and figures to adhere to our mechanistic analysis and avoid any confusions from readers.

To address the issue of egg-laying defects, we quantified total egg number with the mock (*Pocr-3::mCherry*) and PVP^{masc} hermaphrodites. We found that total number of laid eggs was comparable with a slightly higher number in the PVP^{masc} strain, suggesting that egg-laying rate rather fertilization is generally affected.

Altogether, these results are consistent with our main conclusion in which PVP branches engage in egg-laying rate instead of egg-fertilization.

(R) - Many graphs and figure legends throughout the paper do not have error bars or values. This needs to be addressed as is a major issue with rigor.

(A) We thank reviewer for raising this concern. Semi-quantification of neuronal morphology in *C. elegans* often documents the percentage of worms with different shapes/defects in population. This is commonly used in many studies including recent publication for sexually dimorphic PVD branching (Figure 2) and our previous works (Chen *et al*, 2017; Chen *et al*, 2014; Iosilevskii *et al*, 2025). In this case, data are analyzed by the Fish exact test. We also tried to quantify different batches of animals and found that variations are small (as shown in Figure EV2). Thus, we would like to follow the standard in our study.

(R) - The discussion is too extensive and would benefit from focusing on the major findings of the manuscript.

(A) We have modified our discussion and concluded our findings concisely in the first paragraph. To better notify the limitation of traditional transgenes in studying PVP neurons, we added one paragraph for discussion as below:

Line 443-453

Limitations of This Study

Our phenotypic analyses revealed a transgenic toxicity associated with the *ocr-3* promoter (PVP-specific) that affects PVP branch morphogenesis, consistent with previous observations (Christie & Koelle, 2022). Although we minimized this effect by using a relatively low transgene concentration (7.5 ng/ μ L), a mild reduction in PVP branch formation and wing-to-rod transitions was still observed. Consequently, any manipulation of gene activity or expression of synaptic or cytoskeletal markers under the *ocr-3* promoter is likely to partially alter the normal morphology and function of PVP neurons. To address this limitation, future studies could employ CRISPR-mediated insertion of a single-copy transgene driven by the *ocr-3* promoter, or identify alternative PVP-specific promoters with minimal toxicity for labeling and functional assays.

(R) Minor comments:

- The plasticity of the PVP branches is quite an interesting finding. Further experiments such as determining if these branches are present after the worm is done laying most of the eggs (day 5 or 6 of adulthood) might further support this finding.

(A) To address it, we have quantified the D5 adults and found that the ratio of wing-like structure was even increased in D5 adults, suggesting that wing-rod transition is mainly modulated by nutritional status but not sensory feedback of egg-laying circuit.

(R) - The timing of the refeeding experiments (how long after re-feeding was the branching scored?) is not clear in the text/methods. Please clarify.

(A) We put animals for 16-18 hours (overnight) on food before quantification. We have added this information to the Fig. 3A and method section.

In Results (line 178-180):

Notably, these morphological changes were reversible, as re-feeding 16-18 hours with undiluted diets restored the wing-like branches (Fig. 3A).

(R) - Font size is inconsistent throughout the text.

(A) We apologize for the mistake with multiple rounds of editing and we have fixed it.

(R) - Whenever possible images with red/green should be altered to be colorblind friendly. I suggest that at least the heat maps be adjusted to either a single hue or to be grayscale. See <https://www.ascb.org/diversity-equity-and-inclusion/how-to-make-scientific-figures-accessible-to-readers-with-color-blindness/> for examples.

(A) We thank reviewer for this comment. We have modified our heatmap for a double gradient of blue and magenta to improve the visibility for all readers.

(R) - Graph axes would benefit from more precise labeling so they can be interpreted without having to refer to the figure legends (eg Figure S6: Fluorescence Intensity (A.U.) -> TRA-1 Fluorescence Intensity (A.U.)

(A) We have changed the labeling.

(R) Line 51-53: Sentence is not clear.

(A) We have modified the sentence for better clarify as below (line 55-57):

Moreover, peripheral tissues have been shown to promote male-specific muscle and gonadal development via non-autonomous mechanisms by secreted factors in *Drosophila*.

Line 98: Clarify that pdf-1 is a transcriptional reporter

(A) We have modified our expression to clearly indicate the use of the *pdf-1* transcriptional reporter. Changes are shown below (line 100-102):

The transgenic strain carrying *Pocr-3::NeonGreen* specifically labeled PVP neurons, confirmed by co-expression of a transcriptional reporter driven by the *pdf-1* promoter (Fig. EV1A)

(R) Line 197: expressed primarily associated -> expressed and primarily associated

Line 213: TRA-1 is referencing the gene: so it should be tra-1.

Line 214: by a transmembrane protein TRA-2 -> by the transmembrane protein TRA-2

Line 215: by an E3-ubiquitin ligase FEM-3 -> by the E3-ubiquitin ligase FEM-3

Line 512-513: Single hermaphrodite -> A single hermaphrodite

Line 513-514 : Scored egg number -> The number of eggs laid in the plate were scored after 2 hours.

Line 678 and 681: Reference is duplicated

Line 798 and 801: Reference is duplicated

(A) We have corrected all these typos or unclear labeling.

(R) Line 223-224: suggesting that sex identity engages in the insulin... -> Sentence is not clear.

(R) Line 239-240: which induce apoptosis abnormally -> Sentence is not clear.

(R) Line 325-326: Sentence is not clear.

(A) We have modified our expression in these sentences.

References

- Chen CH, He CW, Liao CP, Pan CL (2017) A Wnt-planar polarity pathway instructs neurite branching by restricting F-actin assembly through endosomal signaling. *PLoS genetics* 13: e1006720
- Chen CH, Lee A, Liao CP, Liu YW, Pan CL (2014) RHGF-1/PDZ-RhoGEF and retrograde DLK-1 signaling drive neuronal remodeling on microtubule disassembly. *Proceedings of the National Academy of Sciences of the United States of America* 111: 16568-16573
- Christensen R, de la Torre-Ubieta L, Bonni A, Colon-Ramos DA (2011) A conserved PTEN/FOXO pathway regulates neuronal morphology during *C. elegans* development. *Development* 138: 5257-5267
- Christie NTM, Koelle MR (2022) A neuron that regulates locomotion makes a potential sensory cilium lying over the *C. elegans* egg-laying apparatus. *bioRxiv*: 2022.2009.2019.508547
- Collins KM, Bode A, Fernandez RW, Tanis JE, Brewer JC, Creamer MS, Koelle MR (2016) Activity of the *C. elegans* egg-laying behavior circuit is controlled by competing activation and feedback inhibition. *eLife* 5
- Garcia LR, Portman DS (2016) Neural circuits for sexually dimorphic and sexually divergent behaviors in *Caenorhabditis elegans*. *Curr Opin Neurobiol* 38: 46-52
- Iosilevskii Y, Hall DH, Katz M, Podbilewicz B (2025) The PVD neuron has male-specific structure and mating function in *Caenorhabditis elegans*. *Proceedings of the National Academy of Sciences of the United States of America* 122: e2421376122
- Li C, Chalfie M (1990) Organogenesis in *C. elegans*: positioning of neurons and muscles in the egg-laying system. *Neuron* 4: 681-695
- Mata-Cabana A, Romero-Exposito FJ, Geibel M, Piubeli FA, Mellow M, Olmedo M (2022) Deviations from temporal scaling support a stage-specific regulation for *C. elegans* postembryonic development. *BMC Biol* 20: 94
- Medrano E, Collins KM (2023) Muscle-directed mechanosensory feedback activates egg-laying circuit activity and behavior in *Caenorhabditis elegans*. *Current biology : CB* 33: 2330-2339 e2338
- Yan L, Claman A, Bode A, Collins KM (2025) The *C. elegans* uv1 Neuroendocrine Cells Provide Mechanosensory Feedback of Vulval Opening. *The Journal of neuroscience : the official journal of the Society for Neuroscience* 45

Dear Dr. Chen,

Thank you for the submission of your revised manuscript. We have now received the enclosed reports from the referees and I am happy to say that all support its publication now. Referee 2 has one more comment that I would like you to address in the final ms.

Also a few editorial requests will need to be addressed before we can proceed with the official acceptance of your manuscript:

- Please remove the author credits from the ms file. All credits need to be entered during online ms submission.
- The callouts for the Appendix Figures S1 and S2 are missing an "S"; the Dataset EV1 is called out in the Reagents table in the ms, but the file is missing (may be you are referring to Table EV1?), please correct.
- The legend for Table EV1 needs to be removed from the ms file and provided in the Excel table file itself as a separate sheet or tab.
- 4 Appendix figures are uploaded separately, their legends are in the ms; Appendix Figures and legends need to be provided in a single PDF file titled Appendix (each legend should follow each figure, so they need to be removed from the ms); we also need page numbers and a title page with a table of content and page numbers in the Appendix file.
- Please send us a new synopsis image at the correct size of 550 pixels x 200-600 pixels. The current image at the correct size has blurred text and some text is also too small and not readable.
- The Reagents and Tools table in the ms file needs to be removed, we only need a file that is uploaded separately.
- One Source Data (SD) folder per main figure needs to be uploaded.
- Materials and Methods should be Methods.
- The AI declaration should be moved to the Methods section.
- Main and EV Figure legends should be placed at the end of the ms file.

Figure Legends - Comments

- Please note that the exact p values are not provided in the legends of figures 2E, 3B, C; 5B, D, F; 6C; 7D, EV2 C, S4. Please provide exact values as reasonable.
- Please indicate the statistical test used for data analysis in the legends of figures 2C, E; 5D
- Please note that the error bars are not defined in the legends of figures 2C, E; 3E, 6C, EV3 B.
- Please note that the red arrows are not defined in the legend of figure 1A. This needs to be rectified.
- Please note that the red arrows are not defined in the legend of figure 1D. This needs to be rectified.
- Please note that the blue arrows are not defined in the legend of figure 2F. This needs to be rectified.
- Please note that the blue dotted borders are not defined in the legend of figure 3D. This needs to be rectified.

Referee #1:

The reviewers have done a very nice job in addressing my comments and the manuscript is now acceptable for publication as is.

Referee #2:

The authors addressed most of my concerns. However, I don't accept their reply to my first major concern. Observing dimorphic expression when expressing *daf-16b::mCherry* under the *ocr-3* promoter tells us much more about OCR-3 dimorphic actions than it does about *daf-16* levels in PVP. This means that the authors haven't managed to form a direct link between the sex determination pathway and the regulation of *daf-16*. The results added to lines 219-226 should be removed and instead, the authors should state this in the discussion. The current section added to the discussion (The Interplay Between Biological Sex and the Insulin Pathway) is incorrect and should be phrased as an open point that remains to be answered.

Referee #3:

The authors have adequately addressed all reviewers comments and critiques. The findings and conclusions are well supported by their experimental setups and findings. The manuscript would still benefit from copy editing as there are some typos and issues with sentence construction that should be addressed.

Response to reviewers:

Referee #2:

The authors addressed most of my concerns. However, I don't accept their reply to my first major concern. Observing dimorphic expression when expressing *daf-16b::mCherry* under the *ocr-3* promoter tells us much more about OCR-3 dimorphic actions than it does about *daf-16* levels in PVP. This means that the authors haven't managed to form a direct link between the sex determination pathway and the regulation of *daf-16*. The results added to lines 219-226 should be removed and instead, the authors should state this in the discussion. The current section added to the discussion (The Interplay Between Biological Sex and the Insulin Pathway) is incorrect and should be phrased as an open point that remains to be answered.

We thank the reviewer for the comments and suggestions. We have omitted the related data and mentioned that the direct evidence between sex identity and *daf-16* activity remains unsolved in discussion.

Referee #3:

The authors have adequately addressed all reviewers comments and critiques. The findings and conclusions are well supported by their experimental setups and findings. The manuscript would still benefit from copy editing as there are some typos and issues with sentence construction that should be addressed.

We thank the reviewer for the suggestion. We have corrected typos and modified long sentences in the manuscript.

Dr. Chun-Hao Chen
National Taiwan University
Institute of Molecular and Cellular Biology
No. 1, Sec. 4, Roosevelt Rd.
Taipei
Taiwan

Dear Dr. Chen,

I am very pleased to accept your manuscript for publication in the next available issue of EMBO reports. Thank you for your contribution to our journal.

Yours sincerely,
